# Aminergic and peptidergic modulation of insulin-producing cells in *Drosophila*

Martina Held[1†], Rituja S Bisen[1†], Meet Zandawala[2,3†], Alexander S Chockley[1‡], Isabella S Balles[1§], Selina Hilpert[2], Sander Liessem[1], Federico Cascino-Milani[1], Jan M Ache[1*]

[1]Ache Lab, Neurobiology and Genetics, Theodor-Boveri-Institute, Biocenter, Julius-Maximilians-University of Würzburg, Am Hubland, Würzburg, Germany; [2]Zandawala Lab, Neurobiology and Genetics, Theodor-Boveri-Institute, Biocenter, Julius-Maximilians-University of Würzburg, Am Hubland, Würzburg, Germany; [3]Department of Biochemistry and Molecular Biology, University of Nevada Reno, Reno, United States

*For correspondence:
jan.ache@uni-wuerzburg.de

[†]These authors contributed equally to this work

Present address: [‡]Independent Researcher, Frechen, Germany; [§]Institute of Neurophysiology, Charité–Universitätsmedizin Berlin, Berlin, Germany

## eLife Assessment

This **fundamental** study comprehensively characterizes insulin producing cells (IPCs) resident in the *Drosophila melanogaster* brain. A **compelling** experimental tour de force, the combination of connectomics, mapping of receptors for neuromodulators, electrophysiological recordings, calcium imaging and optogenetics demonstrates that IPCs operate as a functionally heterogeneous population, as necessary to address continuously changing metabolic demands. These findings will be of interest to both neuroscientists and physiologists seeking to study context-dependent neuroendocrine regulation.

**Abstract** Insulin plays a critical role in maintaining metabolic homeostasis. Since metabolic demands are highly dynamic, insulin release needs to be constantly adjusted. These adjustments are mediated by different pathways, most prominently the blood glucose level, but also by feedforward signals from motor circuits and different neuromodulatory systems. Here, we analyze how neuromodulatory inputs control the activity of the main source of insulin in *Drosophila* – a population of insulin-producing cells (IPCs) located in the brain. IPCs are functionally analogous to mammalian pancreatic beta cells, but their location makes them accessible for in vivo recordings in intact animals. We characterized functional inputs to IPCs using single-nucleus RNA sequencing analysis, anatomical receptor expression mapping, connectomics, and an optogenetics-based 'intrinsic pharmacology' approach. Our results show that the IPC population expresses a variety of receptors for neuromodulators and classical neurotransmitters. Interestingly, IPCs exhibit heterogeneous receptor profiles, suggesting that the IPC population can be modulated differentially. This is supported by electrophysiological recordings from IPCs, which we performed while activating different populations of modulatory neurons. Our analysis revealed that some modulatory inputs have heterogeneous effects on the IPC activity, such that they inhibit one subset of IPCs, while exciting another. Monitoring calcium activity across the IPC population uncovered that these heterogeneous responses occur simultaneously. Certain neuromodulatory populations shifted the IPC population activity towards an excited state, while others shifted it towards inhibition. Taken together, we provide a comprehensive, multi-level analysis of neuromodulation in the insulinergic system of *Drosophila*.

## Introduction

Neuromodulation provides the foundation for neural circuit flexibility, which in turn facilitates adaptive behavioral and physiological responses to ever-changing internal demands and environmental conditions (*Marder, 2012*). Neuromodulators, such as neuropeptides and biogenic amines, regulate neural circuits through their diverse effects on excitability and synaptic transmission (*Taghert and Nitabach, 2012*; *Rosikon et al., 2023*; *Nässel and Zandawala, 2022*). Many neural circuits are targeted by multiple neuromodulators, which adds another layer of flexibility while also ensuring stability. Due to a limited number of in vivo model systems to investigate neuromodulation, most studies to date have used ex vivo, or reduced preparations, often investigating one modulatory system at a time (*Marder, 2012*; *Daur et al., 2016*). Here, we investigate the key modulatory inputs to IPCs, a group of approximately 16 neurosecretory cells (*McKim et al., 2024*) in the pars intercerebralis of the *Drosophila* brain (*Cao and Brown, 2001*; *Nässel and Vanden Broeck, 2016*), in vivo. This is timely, since recent advances in the *Drosophila* brain connectome provide an ideal platform to disentangle synaptic connectivity in an organism with a genetically accessible and numerically simple nervous system (*Zheng et al., 2018*; *Scheffer et al., 2020*; *Dorkenwald et al., 2024*). However, the connectome is based on synaptic connectivity and cannot be used to uncover paracrine connections, which are a hallmark of neuromodulation. IPCs are an ideal system to investigate principles of neuromodulation in vivo for several reasons: (1) IPCs release *Drosophila* insulin-like peptides (DILPs), which are analogous to human insulin. The insulin-signaling pathway and its functions are highly conserved across the animal kingdom, from worms to flies to humans (*Nässel and Vanden Broeck, 2016*; *Prentki et al., 2013*). For instance, insulin is essential for many vital processes including metabolic homeostasis, feeding and foraging, aging, and growth (*Cao and Brown, 2001*; *Garofalo, 2002*; *Tatar et al., 2003*; *Fernandez and Torres-Alemán, 2012*; *Zandawala et al., 2018*; *Partridge et al., 2011*; *Nässel et al., 2013*). Hence, IPC activity and the resultant insulin release need to be carefully regulated on different timescales (*Broughton et al., 2005*; *Belgacem and Martin, 2006*; *Nässel, 2012*; *Liessem et al., 2023*). (2) IPCs need to be modulated by different neuronal networks to adjust DILP release according to internal states, metabolic demands, and external inputs (*Nässel et al., 2013*; *Liessem et al., 2023*; *Nässel and Zandawala, 2020*; *Bisen et al., 2024*). (3) IPCs have been identified in the brain connectome (*Reinhard et al., 2023*), facilitating the analysis of their synaptic inputs. (4) The genetic toolkit available in *Drosophila* makes the IPCs not only accessible for in vivo recordings but also allows for simultaneous manipulation of other neuromodulatory populations via optogenetics. The activation of these populations leads to the intrinsic release of neuromodulators in a controlled manner, which we refer to as 'intrinsic pharmacology.' (5) Their superficial location in the brain, in combination with genetic tools, enables the quantification of IPC activity in vivo via approaches like calcium imaging and whole-cell patch-clamp recordings. Here, we combined these approaches to investigate how IPC activity is modulated by functional inputs from a set of core aminergic and peptidergic neuron populations. Thus, our results provide a framework for understanding the regulation and possibly dysregulation of an insulinergic system. In humans, dysregulation of the insulinergic system underlies diabetes, obesity, and other metabolic disorders (*Musselman et al., 2011*; *Ormazabal et al., 2018*).

In adult *Drosophila*, all IPCs express DILP2, 3, and 5 and have traditionally been considered a homogeneous population (*Broughton et al., 2005*). However, emerging evidence has indicated that the IPCs could, in fact, represent a heterogeneous population. Specifically, only a subset of IPCs is intrinsically mechanosensitive via Piezo channels, which are required to prevent food overconsumption (*Wang et al., 2020*). In addition, IPCs are heterogeneous in their responses to the stimulation of clock neurons and their ability to sense glucose (*Oh et al., 2019*; *Barber et al., 2021*). However, how the IPC population and individual IPCs are modulated by different neuronal pathways remains unclear.

In this study, we employed an exhaustive, multi-level approach to leverage *Drosophila* IPCs as a model for investigating neuromodulatory dynamics and principles of metabolic homeostasis in vivo. Using a combination of single-nucleus transcriptome analyses, anatomical receptor mapping, optogenetics, whole-cell patch-clamp recordings, calcium imaging, and connectomics, we unraveled how individual IPCs and the IPC population activity are affected by a broad range of modulatory inputs. We demonstrated that certain modulators shift the overall activity of the IPC population towards excitation or inhibition. This shift could in turn alter the responsiveness of the IPCs, possibly to prime the system for higher or lower metabolic demands. In addition to the effects on the overall population activity, we unveiled that IPCs consist of at least three subpopulations regarding their modulatory input. At the

molecular level, the heterogeneous responses of individual IPCs can partially be explained by differences in their receptor expression profile. We hypothesized that this heterogeneity helps to buffer the system and allows insulin release to be tailored to the ever-changing demands of the animal. Taken together, our work demonstrates that IPCs are part of a complex neuromodulatory network and integrate various regulatory inputs via different pathways. Furthermore, the regulatory motifs identified in our study regarding the orchestration of IPC activity by multiple modulatory systems are likely also present in other modulatory systems, allowing for the generalization of our findings.

## Results

### Single-nucleus transcriptomic analysis identifies novel modulators of IPCs

Previous studies using a combination of molecular techniques and ex vivo recordings have shown that several neuropeptides, biogenic amines, and classical neurotransmitters modulate IPCs (*Nässel and Vanden Broeck, 2016*; *Nässel and Zandawala, 2020*; *Nässel and Zandawala, 2019*). To identify additional modulators of IPCs and confirm previously published findings, we analyzed single-nucleus transcriptomes of IPCs, which were recently sequenced as part of the Fly Cell Atlas project (*Li et al., 2022*). Our analysis was based on 392 IPC transcriptomes, out of which 194 were derived from males and 198 from females. All these cells expressed *Ilp2, 3,* and *5* (*Figure 1—figure supplement 1A*), the best-known marker genes for IPCs. *Drosulfakinin (Dsk)*, which has previously been shown to be expressed in a subset of IPCs (*Wu et al., 2019*; *Söderberg et al., 2012*), was very sparsely expressed in IPCs in this dataset (*Figure 1—figure supplement 1A*). This sparse expression was independently confirmed using a *Dsk-T2A-GAL4* knock-in line (*Figure 1—figure supplement 1B*). Nonetheless, the high expression of all three *Ilp* transcripts in these cells confirmed that these transcriptomes were derived from IPCs.

To catalog all modulators potentially involved in regulating IPC activity, we examined the expression of receptors for all biogenic amines and neuropeptides (*Figure 1A*) in IPC transcriptomes. Using a threshold of 5% expression (see Data analysis for details), we found 22 neuropeptide and 17 biogenic amine receptors to be expressed in IPCs. This included receptors previously shown to be expressed in IPCs like *leucokinin receptor* (*Lkr*), *allatostatin-A receptor 1* (*AstA-R1*), and the serotonin receptor *5-Hydroxytryptamine 1 A* (5-HT1A) (*Nässel and Zandawala, 2020*; *Nässel and Zandawala, 2019*; *Sudhakar et al., 2020*; *Chatterjee and Perrimon, 2021*). In addition, our analysis revealed several novel receptors, including *5-Hydroxytryptamine 7 (5-HT7)* and *5-Hydroxytryptamine 1B (5-HT1B) receptors, rickets (rk), Trissin receptor (TrissinR),* and others (*Figure 1A*). Since there were little differences in receptor expression between males and females (*Figure 1—figure supplement 1C*), we used the transcriptomes from both sexes for all subsequent analyses.

Interestingly, the insulin receptor (*InR*) was the highest expressed neuropeptide receptor in IPCs, suggesting that IPCs are either modulated in an autocrine manner, by DILPs produced by other tissues other than IPCs, or both. Moreover, IPCs express receptors for all classes of biogenic amines (*Figure 1A*). The *Drosophila* genome encodes multiple cognate receptors for all biogenic amines and some neuropeptides. Interestingly, multiple receptors for some of these ligands (e.g. all biogenic amines, Allatostatin-A (AstA), and Diuretic hormone 44 [DH44]) are expressed in IPCs. Since there are functional differences between different receptors for a given ligand (*Larsen et al., 2001*; *Evans and Maqueira, 2005*; *Hector et al., 2009*), co-expression of multiple receptors for a specific ligand in IPCs adds complexity to the modulatory input. In addition, IPCs receive input via fast-acting, small-molecule neurotransmitters since they express receptors for gamma-aminobutyric acid (GABA), glutamate (Glu), and acetylcholine (ACh; *Figure 1—figure supplement 1D*). Taken together, the gene expression analysis indicates that IPCs are intricately regulated by synaptic input via classical neurotransmitter, and paracrine/endocrine input via neuropeptides and biogenic amines.

### IPCs are heterogeneous in their receptor expression profiles

IPCs have generally been regarded as a homogeneous population of approximately 16 cells that can be anatomically identified using antibodies against DILP2, 3 or 5. However, recent evidence suggests that IPCs can be separated into at least two subgroups based on the expression of the mechanosensitive channel Piezo in only a subset of IPCs (*Wang et al., 2020*), heterogeneous responses to

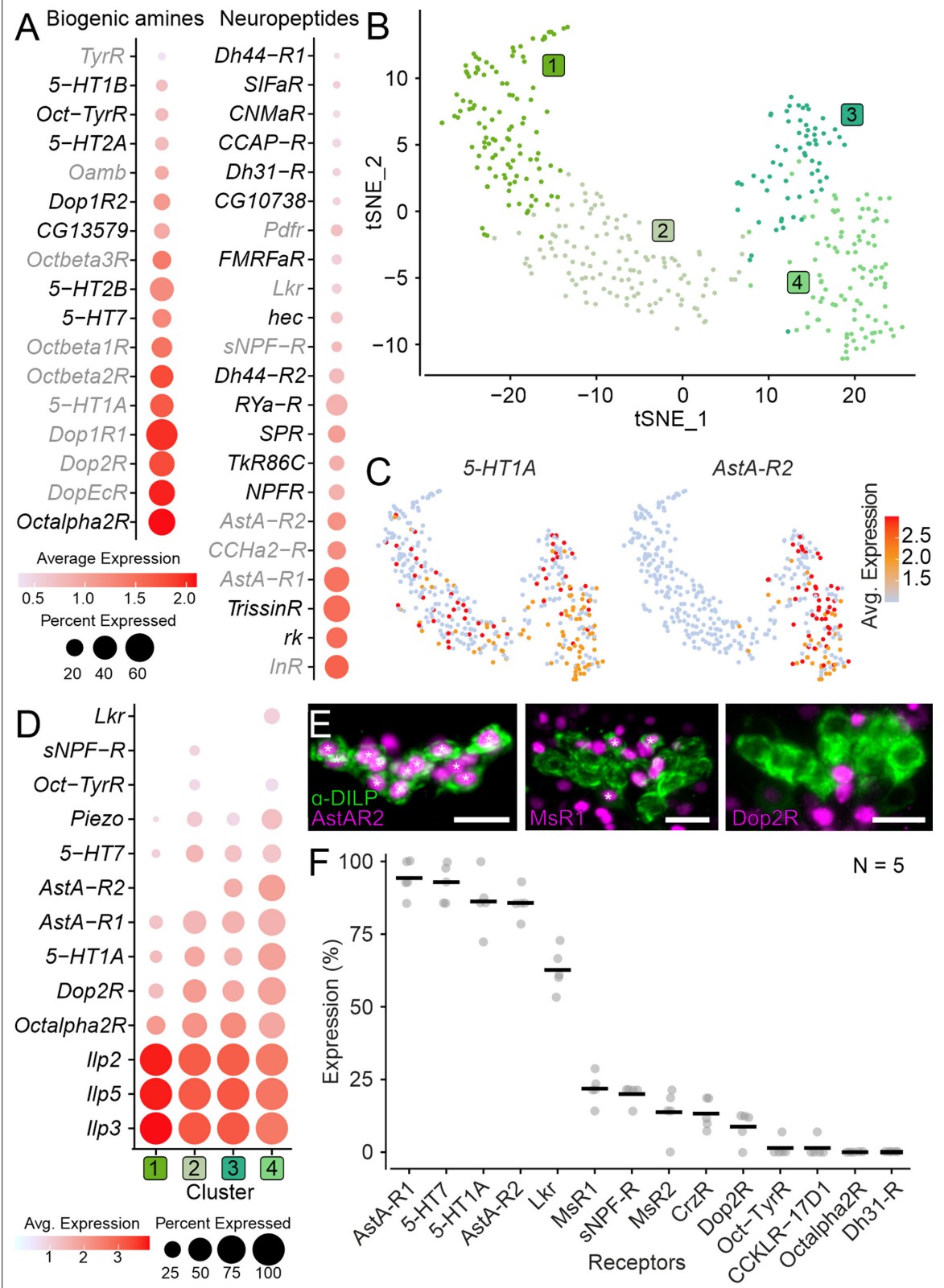

**Figure 1.** Expression of neuromodulator receptors in insulin-producing cells (IPCs). (**A**) Expression of biogenic amine and neuropeptide receptors in IPC single-nucleus transcriptomes. Novel receptors are depicted in black, and receptors previously shown to be expressed in IPCs are shown in gray. (**B**) t-distributed Stochastic Neighbor Embedding (t-SNE) plot based on unsupervised clustering of IPC transcriptomes reveals four clusters. (**C**) t-SNE plots showing expression of 5-HT1A receptor and Allatostatin-A receptor 2. (**D**) Expression of IPC markers (*Ilp2, Ilp3, Ilp5,* and *Piezo*) and select

*Figure 1 continued on next page*

*Figure 1 continued*

neuromodulatory receptors across different IPC clusters. Note that some receptors are expressed in all clusters while others are only expressed in a subset of clusters. A cutoff of 10% expression was used for this analysis. (**E**) Maximum projection of representative confocal images showing receptor-T2A-GAL4 driven mCherry expression (magenta) in IPCs (labeled using DILP2 antibody in green). Asterisks indicate IPCs whose nuclei are stained. Scale bars = 10 μm. (**F**) Fraction of IPCs expressing select receptors based on T2A-GAL4 driven mCherry expression in IPCs. Averages are based on five preparations for each receptor. See *Supplementary file 1A* for all abbreviations.

The online version of this article includes the following figure supplement(s) for figure 1:

**Figure supplement 1.** Insulin-producing cells (IPCs) express receptors for biogenic amines, neuropeptides, and classical neurotransmitters.

**Figure supplement 2.** Expression of select neuromodulator receptors across individual insulin-producing cells (IPCs).

**Figure supplement 3.** Overview of neuromodulator receptor expression in the brain.

**Figure supplement 4.** Neuromodulator receptors expressed in insulin-producing cells (IPCs).

**Figure supplement 5.** Neuromodulator receptors not expressed in insulin-producing cells (IPCs).

clock-neuron stimulation, and their ability to sense glucose (*Oh et al., 2019*; *Barber et al., 2021*). Our analysis confirmed that *Piezo* is only expressed in a subset of IPCs (*Figure 1—figure supplement 1A*). Hence, we analyzed if the receptors expressed in IPCs (*Figure 1A*) were differentially expressed across IPCs, lending further support for the heterogeneity of the IPC population. For this, we performed unsupervised clustering of all IPC transcriptomes based on highly variable genes and visualized the results using t-SNE analysis (*Figure 1B*). This analysis retrieved four clusters that were not very distinct, suggesting a lack of drastic variability in gene expression across IPCs. Nonetheless, examining the expression of 10 neuromodulator receptors that were selected to represent neuropeptides and amines, as well as low and high receptor expression, revealed cluster-specific differences (*Figure 1C and D* and *Figure 1—figure supplement 2*). For instance, *5-HT1A* was expressed in all four clusters, *AstA-R2* was expressed in two clusters (*Figure 1C and D*), while *short neuropeptide F receptor* (*sNPF-R*) was only expressed in one cluster (*Figure 1D*, *Figure 1—figure supplement 2*). Other receptor genes, such as *Piezo*, were expressed across all clusters but at drastically different levels and in different fractions of IPCs. Hence, differences in receptor expression profiles, based on transcriptome sequencing, support the conclusion that the IPCs are a heterogeneous population.

## Anatomical mapping of neuromodulator receptors complements single-nucleus transcriptome profiling of IPCs

Single-nucleus transcriptomes can prove extremely powerful in examining global gene expression differences at a cellular resolution. However, it has limitations, including contamination by ambient RNA, which can give rise to false positives (*Allen et al., 2020*). To assess the reliability of our expression analysis, we verified the expression of 14 neuromodulator receptors in IPCs using anatomical techniques. We primarily used T2A-GAL4 knock-in lines (*Deng et al., 2019*; *Kondo et al., 2020*) to drive expression of nuclear mCherry under the control of endogenous neuromodulator receptor promoters and then quantified the percentage of IPCs expressing mCherry for each receptor line using immunohistochemistry (*Figure 1E and F* and *Figure 1—figure supplement 3*; *Figure 1—figure supplement 4*; *Figure 1—figure supplement 5*). These T2A-GAL4 lines provide insights into which receptor transcripts are translated because the T2A peptides induce ribosome skipping during translation so that the GAL4 protein is only produced when the receptor protein is produced. In accordance with previous anatomical investigations (*Zandawala et al., 2018*; *Kapan et al., 2012*; *Hentze et al., 2015*; *Luo et al., 2012*; *Yurgel et al., 2019*) and our transcriptome analysis, receptors for AstA, LK, sNPF, and 5-HT were all expressed in the IPCs (*Figure 1E and F*; *Figure 1—figure supplement 3* and *Figure 1—figure supplement 4*). Importantly, the expression of several novel neuromodulator receptors identified by our sequencing analysis was also confirmed. For instance, all five serotonin receptor transcripts were detected in the IPC transcriptomes, one of which (*5-HT7*) was further confirmed using anatomical mapping (*Figure 1A and F*, *Figure 1—figure supplement 3* and *Figure 1—figure supplement 4*). These receptor mapping experiments also indicated that not all IPCs are identical, as some receptors were expressed in only a subset of IPCs (*Figure 1*, *Figure 1—figure supplement 4*). For example, AstA-R2 was expressed in 86%, Lkr in 63%, and sNPF-R and myosuppressin (MS) receptor 1 (MsR1) in about 20% of IPCs (*Figure 1F*). Thus, each receptor type is expressed by a different fraction of IPCs, implying that IPCs consist of more than two subgroups. Across individuals,

however, the fractions of IPCs expressing a given receptor were consistent, suggesting that receptor expression is tightly regulated (*Figure 1F*). We used percent expression similar and lower than the corazonin receptor (CrzR) (*Figure 1F*) as the upper bound to determine a receptor as not expressed in IPCs based on anatomical mapping (*Figure 1—figure supplement 5*). This was justified based on the fact that the average CrzR expression (13.3%) was close to the threshold of one pair of IPCs (2 out of 16 or 12.5%) per brain, and previous studies reported a lack of CrzR expression in IPCs (*Oh et al., 2019*; *Zandawala et al., 2021*). Our anatomical mapping largely correlated with our single-nucleus sequencing analysis. Exceptions to this included the two MS receptors, MsR1 and MsR2. T2A-GAL4 lines for both receptors drove expression in a small subset of IPCs but the expression of their transcripts in the IPC transcriptomes was below the threshold used here (*Figure 1*, *Figure 1—figure supplement 3* and *Figure 1—figure supplement 4*). One explanation for the discrepancies could be that transcriptomic analysis provides a single snapshot, whereas anatomical data is based on cumulative expression. Fluorescent markers persist long after transcription and translation have terminated. Therefore, a higher likelihood for receptor expression can be expected when it is quantified via anatomical techniques. Hence, neuromodulator receptors with low expression levels could be expressed in the IPCs in addition to the ones reported here based on the single-nucleus transcriptomic analysis. Other differences between the gene expression analysis and anatomical mapping were exemplified by the dopamine receptor *Dop2R* and the octopamine receptor *Octα2R*, which were both highly expressed in the IPC transcriptomes but were not detected using anatomical mapping (*Figure 1A and F* and *Figure 1—figure supplement 5*). To determine if these discrepancies were due to the presence of additional receptor transcript variants with different C-termini that may not be under the control of the T2A-GAL4 knock-in lines, we utilized a Trojan-GAL4 which should represent expression of other *Octα2R* transcript variants. Surprisingly, this general GAL4 line for *Octα2R* also did not drive expression in IPCs (*Figure 1—figure supplement 5*). Given that G-protein coupled receptors (GPCRs) are usually expressed in low amounts, it is highly unlikely that these receptors represent false positives due to ambient RNA contamination during sample preparation for transcriptome sequencing (*Sriram et al., 2019*). Hence, inadequate transgenes for *Dop2R* and *Octα2R* or the lack of protein translation are the likely cause for the discrepancy between transcriptome analysis and anatomical mapping.

In summary, anatomical receptor mapping complemented the single-nucleus transcriptome analysis and confirmed the heterogeneous expression of several novel neuromodulator receptors in IPCs. Using two different methods to assess expression of receptors in IPCs also demonstrated the necessity of parallel approaches to obtain accurate insights into receptor expression in complex modulatory systems.

## IPC activity is modulated by aminergic and peptidergic neurons

Previous studies and our receptor expression data suggested that the IPC activity could be modulated by several different modulatory systems acting in parallel, the combined activity of which could ultimately determine insulin release. However, the functional significance of these pathways remains unclear. Therefore, we next sought to investigate how the activation of specific modulatory populations affects IPC activity in vivo. To this end, we recorded the activity of individual IPCs via whole-cell patch-clamp while optogenetically activating different populations of modulatory neurons (ModNs) via CsChrimson (*Figure 2A and B*). We used nine different driver lines targeting different populations of ModNs for the activation experiments. Receptors for most of the neuromodulators found in these neurons were expressed in IPCs according to our expression analyses. We further included a driver line for neurons expressing adipokinetic hormone (AKH), since it was previously shown to influence DILP3 release in larvae (*Kim and Neufeld, 2015*). Hence, all nine of these ModNs have been suggested to affect IPC activity or play a role in metabolic homeostasis and feeding (*Nässel and Vanden Broeck, 2016*; *Zandawala et al., 2018*; *Nässel and Zandawala, 2020*; *Oh et al., 2019*; *Nässel and Zandawala, 2019*; *Yurgel et al., 2019*; *Lin et al., 2022*; *Luo et al., 2014*; *Birse et al., 2011*; *Gough et al., 2017*; *Hallier et al., 2016*; *Takeda et al., 2018*). These specific ModN driver lines targeted neurons containing the biogenic amines dopamine (DANs), octopamine (OANs), or serotonin (5-HTNs), as well as the neuropeptides AstA (AstANs), AKH (AKHNs), diuretic hormone 31 (DH31Ns), LK (LKNs), MS (MSNs), or tachykinin (TKNs, *Figure 2—figure supplement 1*). We also included a line we expected to target sNPF-expressing neurons, but for which anatomical labeling

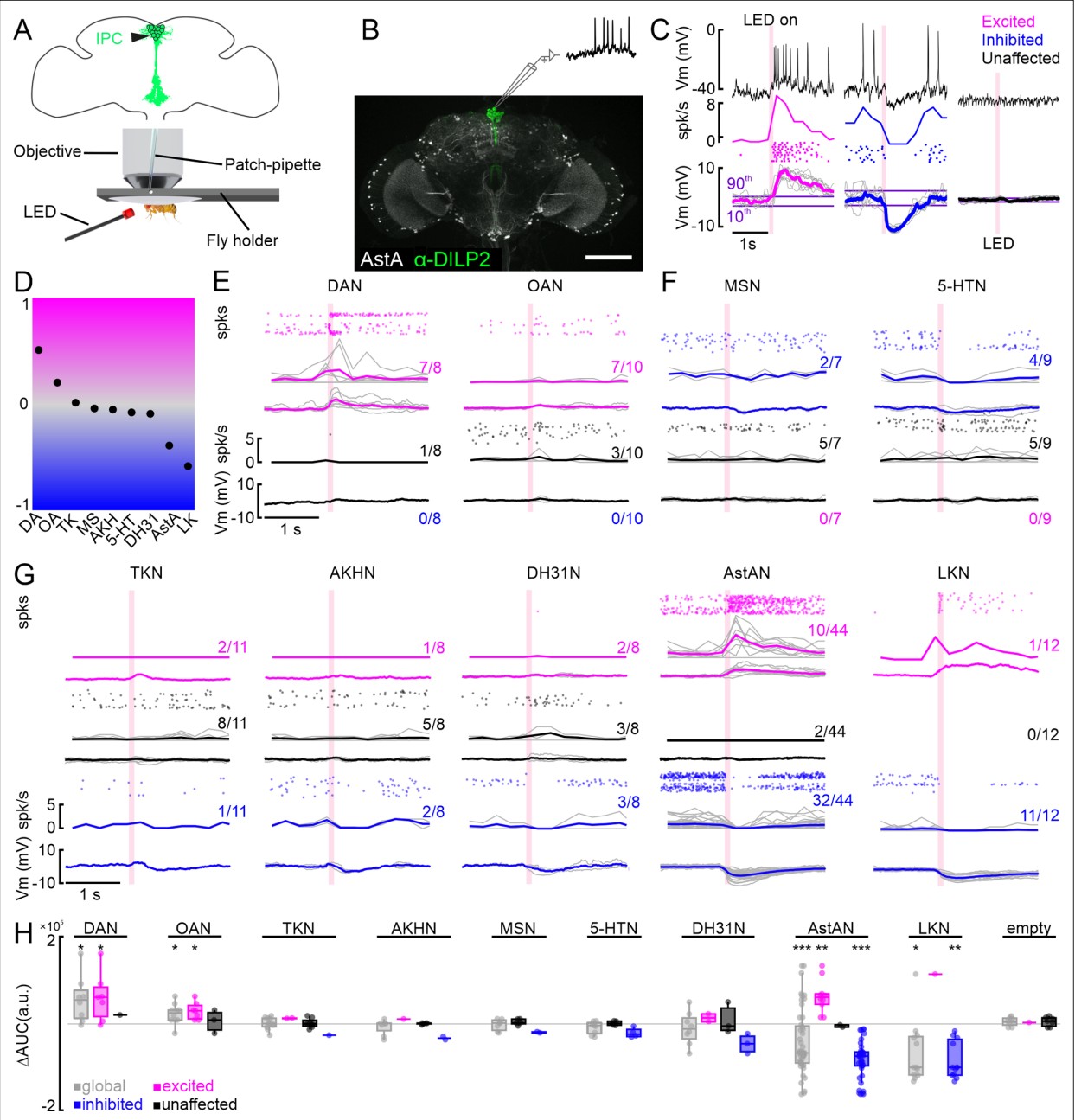

**Figure 2.** Modulation of individual insulin-producing cells (IPCs) by aminergic and peptidergic neurons in patch-clamp recordings. (**A**) Schematic of setup for patch-clamp recordings and optogenetic activation. (**B**) Anatomy of an example driver line labeling AstANs (gray) and antibody staining against DILP2 labeling IPCs (green). (**C**) Example responses of three IPCs to optogenetic activation of AstANs (red bar). IPCs with a membrane potential (Vm) rising above the 90th percentile of the baseline during or after stimulation were defined as 'excited' (magenta). IPCs with Vm falling below the 10th percentile were defined as 'inhibited' (blue). IPCs remaining between the thresholds were 'unaffected' (black). (**D**) Normalized overall trends across all tested lines based on the area under the curve (AUC) average of all recorded cells for the respective line. (**E**) DAN, and OAN populations primarily drove excitation or had no effect on IPCs upon activation. Spike events (dots), spike frequency (spk/s), and low-pass filtered Vm (mV) are shown, color-coded according to the clusters as before. Gray lines indicate individual IPCs, while color-coded lines show the cluster average. Fractions indicate number of IPCs per cluster. (**F**) MS- and 5-HT-expressing neurons had inhibitory or no effects on IPCs. Details as in D. (**G**) TKNs, AKHNs, DH31Ns, AstANs, and LKNs evoked mixed effects in IPCs upon activation. Details as in D. (**H**) Baseline subtracted AUC values (ΔAUC) of all recorded cells (gray), and excited (magenta), unaffected (black), and inhibited (blue) clusters. p-values were calculated using the Wilcoxon signed-rank test.

The online version of this article includes the following figure supplement(s) for figure 2:

**Figure supplement 1.** Expression of the ModN driver lines in the brain and VNC.

**Figure supplement 2.** Insulin-producing cell (IPC) responses to repeated activation of ModNs and statistical analysis of IPC responses to ModN activation for all lines.

revealed that the line drove no expression in the brain or VNC. This line was therefore 'empty', and lent itself as a negative control (*Figure 2—figure supplement 1* and *Figure 2—figure supplement 2D*). While some of these modulators are expressed and released by large populations of ModNs forming distinct sub-populations (e.g. AstA, DA, and 5-HT), others are expressed in very few, individually identifiable neurons (e.g. LK and AKH). Accordingly, the number of targeted neurons varied between driver lines (*Figure 2—figure supplement 1*).

During these intrinsic pharmacology experiments, we first recorded the baseline activity of IPCs, which had a membrane potential of about –40 mV (without subtraction of the 13 mV liquid junction potential *Gouwens and Wilson, 2009*). The spike frequency was 0.8 Hz on average, in flies fed ad libitum. Since baseline recordings from IPCs, in addition to our transcriptomic analysis, revealed no significant difference between male and female flies (*Bisen et al., 2024*), we used mated females for our physiological experiments. Activating individual ModN populations had strong and immediate effects on IPC activity in several cases (*Figure 2C*). To classify IPC responses using a data-driven approach, we implemented threshold criteria for the membrane potential to cluster the IPCs depending on the effects of ModN activation into excited, inhibited, or unaffected (*Figure 2C*, for details see Material and methods). Excitation was characterized by a fast depolarization of the membrane potential, often accompanied by an increase in spike frequency shortly after the onset of activation (*Figure 2C*, magenta). Opposite to this, inhibition was characterized by a hyperpolarization of the membrane potential and a reduction in spike frequency (*Figure 2C*, blue). Unaffected IPCs displayed no significant changes after activation compared to the baseline (*Figure 2C*, black). For each recorded IPC, we activated one ModN population repeatedly for ten trials. Individual IPCs showed consistent responses to repeated activation (*Figure 2C*, *Figure 2—figure supplement 2A and B*). Using this approach, we quantified and categorized the effect of each ModN population on the IPC activity. We then analyzed the global effect of each ModN population on the IPC population activity. To this end, we calculated a general effect coefficient based on the mean area under the curve (AUC) of the membrane potential of all recorded IPCs after ModN activation (see Material and methods for calculation details), describing a global trend for the shift in IPC activity (*Figure 2D*). This coefficient was highest for DAN activation (0.51), meaning the overall activity of the IPCs was strongly shifted towards excitation. OAN activation also led to a positive coefficient (0.2), indicating a weaker excitatory shift. Activation of TKNs (0.01) and MSNs (–0.04) both resulted in a coefficient near zero and, therefore, did not seem to shift the overall activity. AKHNs (–0.05), 5-HTNs (–0.08), and DH31Ns (–0.09) shifted the activity of the IPC activity slightly towards inhibition. Activating AstANs (–0.39) had a stronger net inhibitory effect, and the activation of LKNs (–0.58) led to the strongest inhibition of the IPC population.

Next, we analyzed the effects of each ModN population on individual IPCs in detail. Activation of DANs and OANs either excited or had no effect on individual IPCs (*Figure 2E*). DAN activation resulted in a strong excitation in the majority of IPCs (7/8), and only 1/8 IPCs remained unaffected. Activation of OANs also strongly excited most IPCs (7/10), while 3/10 IPCs were unaffected (*Figure 2E*). In contrast, activation of MSNs and 5-HTNs either inhibited or had no effect on individual IPCs (*Figure 2F*). In one IPC, a single spike was reliably elicited upon activation of 5-HTNs, but the average membrane potential did not meet our threshold and, therefore, it was not classified as excited. Activation of MSNs led to the inhibition of 2/7 IPCs, while 5/7 remained unaffected. Activating 5-HTNs inhibited 4/9 IPCs and had no effect on 5/9 (*Figure 2F*). Surprisingly, activating AstANs, AKHNs, DH31Ns, LKNs, or TKNs, had distinct and even opposite effects on different subsets of IPCs. Here, we found IPCs exhibiting all three response types (excited, inhibited, and unaffected) during activation of the same ModNs (*Figure 2G*). When activating TKNs, 2/11 IPCs were excited, 1/11 inhibited, and the majority, 8/11, remained unaffected. With the activation of AKHNs, only 1/8 IPCs was excited and 2/8 inhibited, while 5/8 remained unaffected. Activating DH31Ns had a stronger effect than AKHNs, with 2/8 IPCs excited, 3/8 inhibited, and only 3/8 remaining unaffected. Interestingly, AstAN activation resulted in strong excitation in one subset, and strong inhibition in another subset of IPCs. Therefore, we decided to analyze the effects of AstANs on IPCs in more detail by increasing the sample size. We found that 10/44 recorded IPCs were excited upon activation, while the largest subset with 32/44 were inhibited, and only 2/44 remained unaffected. LKNs showed an even stronger inhibitory effect, since their activation led to only 1/12 excited IPCs, while 11/12 were inhibited. We also recorded from IPCs while activating the line that was expected to express CsChrimson in sNPFNs but was in fact empty (*Figure 2—figure supplement 1* and *Figure 2—figure supplement 2D*). Accordingly,

'activating' this line while recording from IPCs had no effect on the IPC activity, except in one cell where the membrane potential increased slightly (*Figure 2—figure supplement 2D*). To analyze the effect strength for all lines, we subtracted the baseline AUC, calculated before the activation onset, from the AUC after activation (ΔAUC). To determine if the effects were significant for the respective line, we calculated the p-values using the Wilcoxon signed-rank test for all recorded IPCs and for individual clusters with a minimum number of five recorded IPCs. Our analysis revealed that only the effects of DANs, OANs, AstANs, and LKNs were significant (*Figure 2H*). For DANs and OANs, we found a significant effect for the population of all recorded IPCs and in the excited cluster. For LKNs, on the other hand, the IPC population and the inhibited cluster showed a significant effect. AstAN activation led to a significant effect in the IPC population, in the excited and the inhibited cluster. TKNs, AKHNs, MSNs, 5-HTNs, and DH31Ns also had effects on the IPC activity, especially in the clusters we found for the respective line, but they were not statistically significant. Taken together, these results show that different populations of ModNs have distinct effects on the IPC activity. The heterogeneous effects the activation of individual ModN populations had on IPCs could arise from the heterogeneous receptor profiles exhibited by individual neurons in the IPC population. In addition, our intrinsic pharmacology approach does not only drive neuromodulator release but also the release of classical, fast-acting transmitters contained in most ModNs. Therefore, IPC responses could be driven both by the respective neuromodulator and the co-release of fast-acting neurotransmitters present in those neurons. This co-transmitter release occurs under naturalistic circumstances (*Croset et al., 2018*), and our IPC transcriptome analysis showed that IPCs express receptors for the classical transmitters ACh, Glu, and GABA (*Figure 1—figure supplement 1D*).

The high temporal resolution of our in vivo patch-clamp recordings enabled us to analyze the detailed temporal dynamics of IPC modulation. Here, we focused on DANs, OANs, AstANs, and LKNs, which had strong, significant effects on the IPC activity (*Figure 2H*). We quantified the latency from activation onset to IPC de- or hyperpolarization for each ModN population (*Figure 3A*). All ModNs had rapid effects on the IPC activity, with DANs driving the fastest excitatory effects with a median latency of 10 ms, while OANs had the slowest with a median latency of 122 ms. Inhibitory effects of AstAN and LKN activation occurred with a similar median latency of 49 ms and 43 ms, respectively. Classical monosynaptic connections have a response latency of a few milliseconds (*Griffith, 2012*), which is much faster than the latencies we observed, even when taking into account that it will take a few milliseconds to depolarize the presynaptic neurons optogenetically. We, therefore, assume that the majority of the effects we observed were not due to direct, monosynaptic input to the IPCs. Instead, they are likely driven by polysynaptic or modulatory connections. Only the excitatory effects of DANs and the single excited IPC during LKN activation were on a timescale that could indicate monosynaptic connections.

## Heterogeneous IPC responses occur within the same animal

The heterogeneity we observed in the functional connectivity between ModNs and individual IPCs during patch-clamp recordings could have three different origins. (1) It could reflect differences in synaptic connectivity between different animals. (2) Functional connectivity between ModN populations and IPCs could change over time. For example, the heterogeneity could arise from changes in the animal's internal state, including starvation or changes in behavior, such as increased leg flailing, which readily occurs during in vivo recordings. We controlled for some, but not all these aspects. For example, we provided food to all animals ad libitum until we prepared them for experiments, and we continuously perfused the brain with sugar-containing saline. (3) The heterogeneity could be a persistent feature of IPC modulation, allowing for a more nuanced shift of IPC population activity through the action of ModNs on individual IPCs. This nuanced modulation might be beneficial considering that DILP release is critical for metabolic homeostasis and needs to be flexibly adjusted to varying demand with a relatively small population of IPCs.

To determine if the heterogeneity was a result of inter-individual differences in connectivity and/or internal states, we tested whether IPC responses to ModN activation were less heterogeneous when recording from multiple IPCs in a single animal, rather than inferring the population activity from recordings across animals. We, therefore, subsequently recorded from 11 IPCs in the same fly while activating AstANs. Remarkably, responses across the population of IPCs recorded successively in the same fly were just as heterogeneous as responses recorded across flies: 2/11 IPCs were excited,

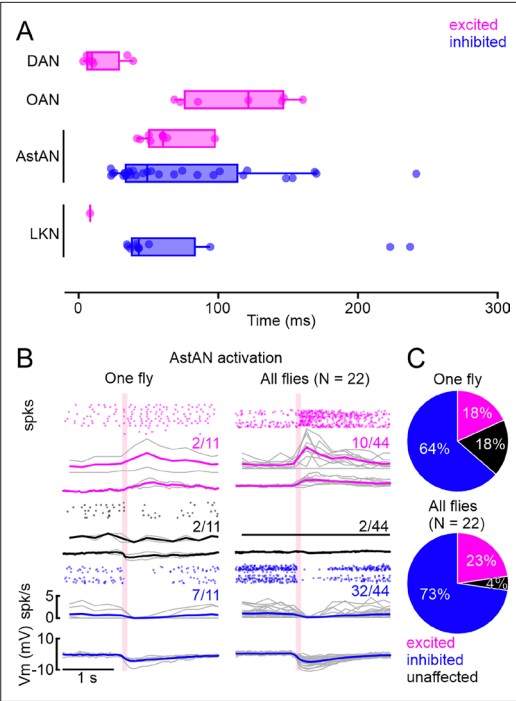

**Figure 3.** Latency analysis and singly fly recordings. (**A**) Threshold-based latency analysis for excited (magenta) and inhibited (blue) insulin-producing cells (IPCs) during activation of dopamine (DANs), octopamine (OANs), neuropeptides AstA (AstANs), and LKNs, which had significant effects on the IPC activity upon activation. Activation onset at 0 s. Boxplots show median and interquartile range, dots represent individual recordings. Three outliers for AstAN between 0.4 and 0.8 not shown for clarity of inspection. (**B**) Patch-clamp recordings of 11 IPCs subsequently patched in the same fly reveal a similar distribution of excited, inhibited, and unaffected IPCs compared to recorded IPCs across all flies (Plot from *Figure 2G*.). (**C**) Cluster distribution across 11 cells in one fly and in 44 cells in 22 flies.

The online version of this article includes the following figure supplement(s) for figure 3:

**Figure supplement 1.** Cluster distribution of multiple insulin-producing cells (IPCs) recorded in the same individuals during neuropeptides AstA (AstANs) activation.

7/11 inhibited, and 2/11 remained unaffected in the same fly (*Figure 3B*). These heterogeneous responses and their distribution in a single animal mirrored the responses of IPCs recorded across multiple animals (*Figure 3C*). Recording the activity of multiple IPCs in four additional animals showed similarly heterogeneous response profiles (*Figure 3—figure supplement 1*). Hence, heterogeneous IPC responses are not accounted for by inter-individual differences. Instead, the heterogeneity occurred within single animals and appears to be a persistent feature of IPC modulation.

## Modulation of individual IPCs underlies shifts in population activity

Our recordings from multiple IPCs in the same animal revealed that the heterogeneous responses were not based on functional differences between individuals. In principle, however, the heterogeneity could still arise from changes in connectivity over time, for example, due to internal state changes. Therefore, we next recorded from multiple IPCs simultaneously via in vivo widefield calcium imaging, to address whether excitatory and inhibitory responses were elicited at the same time in the same animal and to gain a better understanding of the IPC population dynamics (*Figure 4A*). To this end, we recorded the calcium dynamics in IPCs during the activation of DANs, OANs, and MSNs. We chose these lines since they generated the full spectrum of responses from primarily excitatory to inhibitory in patch-clamp recordings. In addition, we selected lines targeting AstANs and LKNs, which elicited heterogeneous effects in our patch-clamp recordings. We tested different activation durations to account for the slower dynamics of the calcium indicator (*Figure 4—figure supplement 1A*). A 5 s activation seemed most appropriate, since it elicited consistent IPC responses while ensuring that IPCs retained naturalistic activity patterns throughout the duration of the experiment (*Figure 4A–C*, *Figure 4—figure supplement 1A and B*). Activation of the selected driver lines evoked strong and immediate fluorescence changes across the IPC population (*Figure 4C*). When comparing IPC responses during two consecutive activations, we found that the same subset of IPCs showed a fluorescence increase during both activations while other IPCs consistently showed a decrease (*Figure 4D*). Hence, individual IPC responses were consistent across trials, just like in our electrophysiology experiments. Therefore, we performed a threshold-based analysis to cluster the responses of individual IPCs into excited, inhibited, and unaffected, analogous to the approach used for electrophysiology data (*Figure 4E*, see for details Material and methods). Imaging multiple IPCs simultaneously revealed that the three activity clusters occurred within the same animal, and at the same time (*Figure 4E*). We visualized the ΔF/F traces for each recorded IPC for all animals in the respective ModN line as heatmaps (*Figure 4F*).

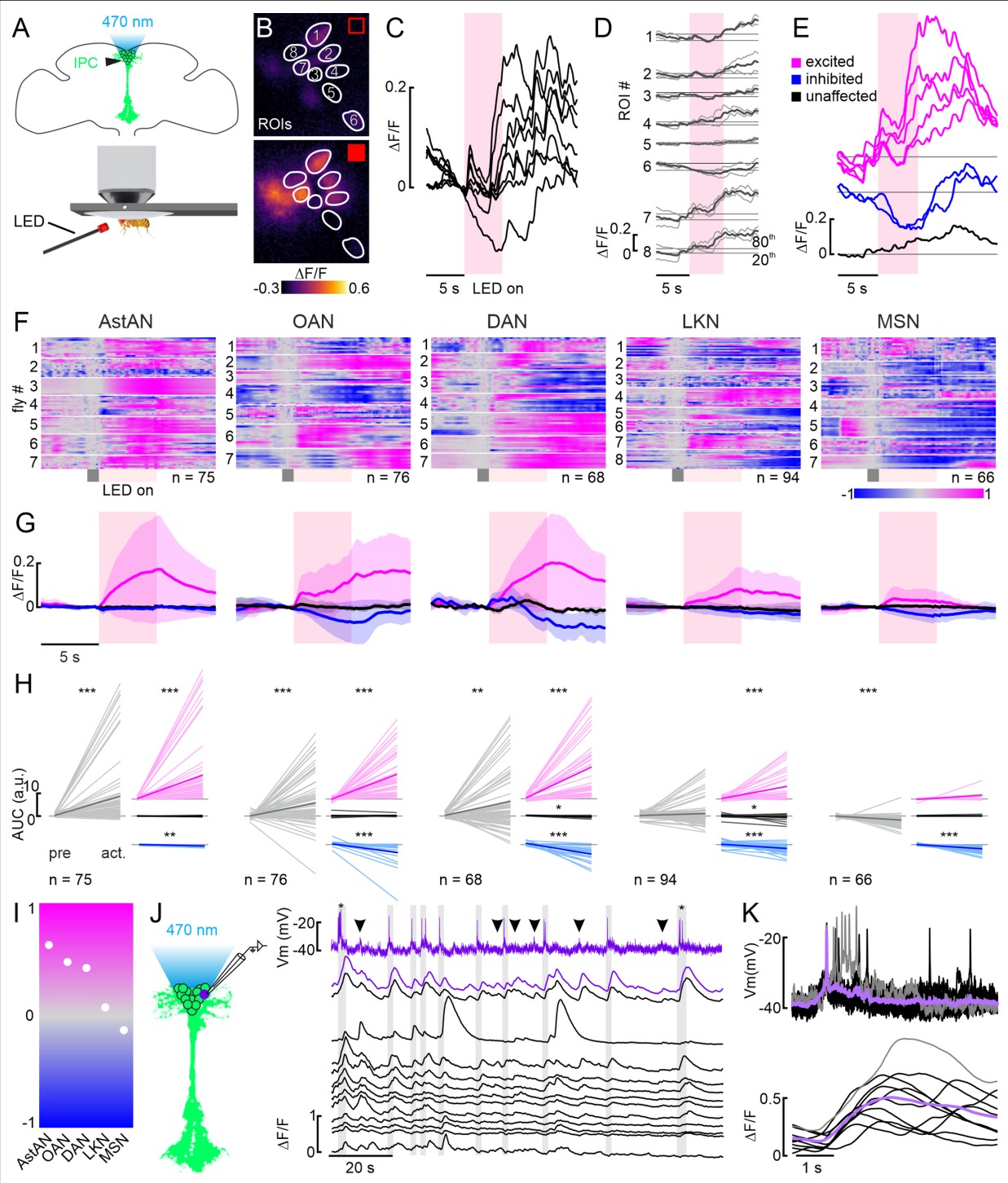

**Figure 4.** Modulation of the insulin-producing cell (IPC) population by aminergic and peptidergic neurons in calcium imaging experiments. (**A**) Schematic of the setup for optogenetic activation during calcium imaging. (**B**) Example images of the GCaMP6m-expressing IPC cell bodies, delimited as ROIs during a time series before and during a 5 s optogenetic activation of octopamine (OANs) (**C**) Normalized ΔF/F traces for regions of interests (ROIs) in B during one activation (red shading). (**D**) Superimposition of two subsequent activations (thin lines) and the mean of both (thick line) from the example in B. The mean was used for further analysis steps. (**E**) Cluster analysis based on the 20th and 80th percentile of the baseline (see Material and methods) revealed three cluster within this one example animal. (**F**) Heatmaps of the mean ΔF/F traces of all recorded IPCs per driver line. Traces were baseline subtracted based on the activity during a 1 s window (gray box) before the 5 s activation (red box). (**G**) Mean ΔF/F traces (thick lines) for each cluster (color-coded as before) with respective standard deviation (shaded areas). Data correspond to the respective driver line indicated in F. (**H**) Area under the curve (AUC) of IPCs before activation (pre) and during activation (act.; from onset to 5 s after activation) for all recorded IPCs (gray, left) and

*Figure 4 continued on next page*

*Figure 4 continued*

for the three clusters (colors, right). p-values were calculated using the Wilcoxon signed-rank test (see *Table 1* for all p-values). (**I**) Overall trends across all tested lines based on the AUC averaged across all IPC responses. (**J**) Simultaneous recording of one IPC via patch-clamp (first row, purple), and the same IPC (second row, purple ΔF/F trace) as well as 12 additional IPCs (black ΔF/F traces) via calcium imaging. Gray boxes indicate action potentials, asterisks indicate bursts of action potentials, and arrowheads point at sub-threshold membrane potential changes leading to calcium increases. (**K**) Superimposition of action potentials from J and corresponding ΔF/F traces (black traces, individual events, lavender trace, mean, gray trace, burst of action potentials, and respective ΔF/F trace).

The online version of this article includes the following figure supplement(s) for figure 4:

**Figure supplement 1.** Comparison of insulin-producing cell (IPC) responses to ModN activation for different durations in calcium imaging recordings.

These high-resolution heatmaps depict the effects of the respective ModN activation on all recorded IPCs, across all animals. For example, the activation of AstANs led to a fluorescence increase in the majority of IPCs across all animals, while the activation of MSNs decreased the fluorescence in most IPCs in all tested animals (*Figure 4F*). However, we also found differences between individual animals. For example, during the activation of OANs, IPCs in flies #2, 5, and 6 were primarily excited, while IPCs in fly #4 were primarily inhibited. Interestingly, we found heterogeneous IPC responses across all driver lines, independent of the strength or valence of the global trend of the IPC population (*Figure 4F*). Recording from multiple IPCs at the same time uncovered that the activity of IPCs within a cluster was synchronized in some cases. For example, in fly #1 in the DAN activation experiment, the baseline activity pattern of the excited IPC cluster was already synchronized before the first activation (fly #1, cells 3–8). Furthermore, the excitation onset and duration during the activation of DANs was highly uniform in this cluster. However, in other flies, e.g., #2 and #3 in the DAN activation experiments, we did not observe this synchronicity. While all IPCs in the excited cluster displayed an excitatory response to the DAN activation in these flies, the onset and duration differed between individual IPCs. In addition, the IPCs also showed more variability in their baseline activity (*Figure 4F*). These findings point towards a shared input that can lead to the synchronization of IPC activity in some clusters and time windows. One known such input is the behavioral state – flight strongly inhibits the activity of all IPCs with very short delays (*Liessem et al., 2023*). The flies in our experiments were not flying, but this example illustrates the presence of strong, state-dependent inputs that can synchronize the IPC population activity. The threshold-based cluster analysis revealed three clusters based on the responses of the IPCs to the activation of the respective driver line for all ModNs tested (*Figure 4G*). To further quantify the effects of ModNs on IPCs and to assess whether they were statistically significant, we compared the activity of each IPC before activation to the activity after activation based on the AUC of individual ΔF/F traces (*Figure 4H*). First, we performed this analysis across the entire IPC population for each driver line. Second, we analyzed the IPCs by cluster, i.e., the excited, inhibited, and unaffected IPCs separately. AstANs, OANs, and DANs had significant overall effects on the population activity, driving a net increase in IPC activity (*Figure 4H*, gray). For these ModNs, the effects were also significant when considering the inhibited and excited cluster separately (*Figure 4H*, blue and magenta). MSN activation significantly inhibited the IPC population activity, an effect that was carried solely by a significant effect on the inhibited IPC cluster (*Figure 4H*, blue). LKN activation had significant effects in the excited and inhibited clusters, but those canceled each other out in the pooled data, resulting in no net excitation or inhibition (*Figure 4H*, gray). The global effect of each ModN population on the IPCs is summarized in *Figure 4I*. This analysis demonstrated the effect strength during the activation of the respective ModNs, with AstANs, OANs, and DANs shifting the population towards an excited state, while MSNs shifted the population activity towards inhibition (*Figure 4I*). These shifts likely change the responsiveness of the IPCs to additional inputs, with AstAN, OAN, and DAN input increasing the responsiveness of the system, while MSN dampens it.

In addition to the population-level analysis, calcium imaging allows the investigation of receptor-dependent calcium dynamics that are not coupled to electrical activity. For example, intracellular calcium mediates changes in gene expression in addition to transmitter release (*Brini et al., 2014*), which might underlie long-term changes in IPC activity. Our intrinsic pharmacology experiments during calcium imaging revealed that activating AstANs and LKNs caused calcium dynamics that were not necessarily indicated by the electrophysiological data. For example, in both cases, we observed strong increases in calcium in the IPCs, which occurred in addition to the inhibition recorded via patch-clamp. Especially AstAN activation resulted in a fast and robust calcium influx across the majority of

recorded IPCs. This calcium increase could lead to an increase in DILP release, but also to alterations in gene transcription, for example. To investigate how the electrophysiological activity recorded in patch-clamp translates into calcium signals in IPCs, we recorded the spontaneous baseline activity from one IPC via patch-clamp while performing calcium imaging of 13 IPC cell bodies (regions of interest [ROIs]) without activating any ModNs. When comparing the membrane potential with the fluorescence traces of the same IPC, fluorescence increases matched the occurrence of individual action potentials faithfully (*Figure 4J*, gray boxes and purple traces). Accordingly, bursts of action potentials led to a prolonged and stronger calcium influx (*Figure 4J* asterisks). By superimposing the action potentials and the associated fluorescence traces, the close correlation became even more apparent (*Figure 4K*), with the expected delay for calcium imaging rooted in the kinetics of GCaMP6m (*Chen et al., 2013*). However, when inspecting the membrane potential and the fluorescence trace of the patched IPC closely, we noticed that in some instances calcium influxes were not driven by action potentials, but by subthreshold depolarization (*Figure 4J*, arrowheads). Hence, while there was a strong correlation between IPC spiking and calcium influx, calcium dynamics were not solely coupled to action potentials. Interestingly, when comparing the fluorescence traces of all 13 IPCs with each other, we found that the fluorescence dynamics were synchronized between some, but not all IPCs (*Figure 4J*). This suggests that some IPCs either receive the same input or form reciprocal connections, while others receive different inputs and are not coupled to the same sub-population. This further demonstrates that IPCs form distinct functional clusters. However, we found no evidence in our anatomical data, calcium imaging experiments, or in the fly brain EM volume that these clusters are distinguishable based on IPC soma location in the pars intercerebralis.

## IPCs receive heterogeneous synaptic input

After investigating the functional inputs to IPCs, we next analyzed direct synaptic connections. To this end, we used the FlyWire whole-brain connectome (*Zheng et al., 2018*; *Dorkenwald et al., 2024*; *Schlegel et al., 2023*; *Eckstein et al., 2023*; *Lin et al., 2024*) to identify presynaptic neurons that connect to IPCs (*McKim et al., 2024*; *Reinhard et al., 2023*) via at least five synapses (see Materials & methods for more details). These neurons were predicted to contain either ACh, Glu, GABA, or unknown transmitters (*Figure 5A-D*). 'Unknown' refers to neurons for which transmitter prediction scores were low, and therefore inconclusive (*Eckstein et al., 2023*), which are primarily ModNs. We grouped the presynaptic neurons according to their neurotransmitter identity and calculated the overall strength of the synaptic input each IPC receives from neurons of each group (*Figure 5E*). This analysis revealed that the IPCs receive heterogeneous inputs from neurons expressing three classical transmitters. Hence, IPCs receive heterogeneous synaptic as well as modulatory inputs. This heterogeneity is most apparent in GABAergic connections, since only eight IPCs receive inputs from GABAergic neurons. Presynaptic neurons containing ACh or Glu provide input to all IPCs. However, the synaptic strength of the individual connections was heterogeneous, ranging from 7 synapses to 63 for cholinergic and 8–36 synapses for glutamatergic inputs. Even though we cannot define the transmitters released by neurons in the 'unknown' category, their connectivity to individual IPCs also varied strongly. The heterogeneity in synaptic input likely contributes to the heterogeneous responses of the IPC population we observed in functional connectivity experiments.

Taken together, our intrinsic pharmacology experiments showed that the different modulatory populations had immediate effects on IPC activity. In calcium imaging experiments, the overall trends of the IPC population activity ranged from a strong calcium increase, and hence excitation, during the activation of AstANs, to a strong decrease in calcium, indicating inhibition, during the activation of MSNs (*Figure 6A*). The activation of DANs, OANs, and MSNs led to global shifts in the population activity that were similar in calcium imaging and patch-clamp experiments. DANs and OANs had a strong excitatory effect, while MSNs had a slightly inhibitory effect. For AstANs and LKNs, calcium imaging revealed different activity dynamics than patch-clamp recordings. In patch-clamp recordings, AstANs had a strong inhibitory effect on the majority of IPCs and a strong excitatory effect on a smaller subset. In contrast, in calcium imaging experiments, AstAN activation led to a strong calcium increase in the majority of IPCs. LKN activation also had a stronger inhibitory effect in patch-clamp recordings than in calcium imaging experiments, which is apparent when comparing the global effects of both lines between imaging and patch-clamp experiments (*Figure 6A*). These similarities and differences in the overall effect were not only driven by the effect strength but also

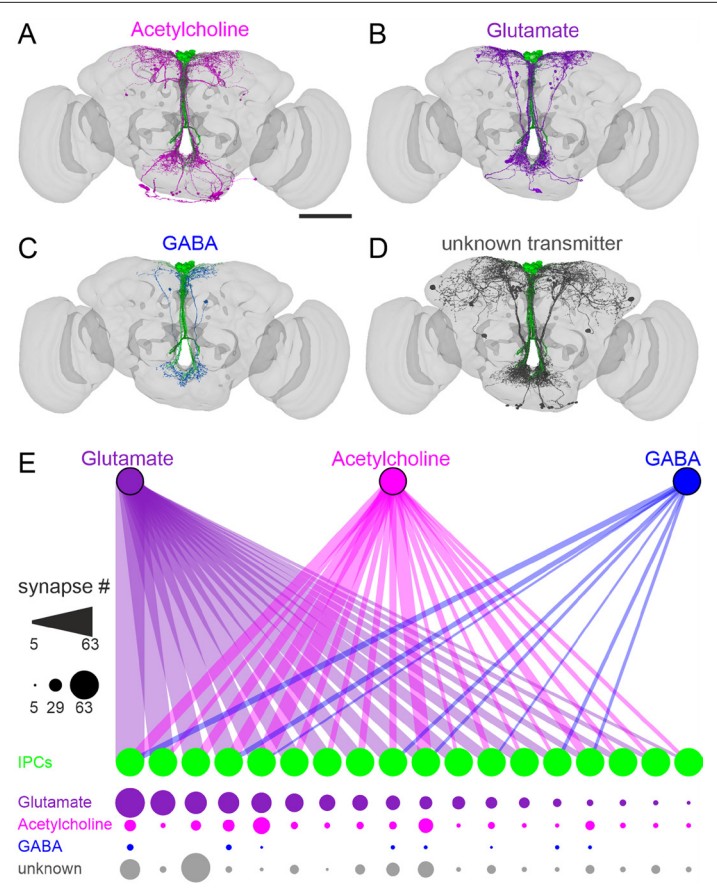

**Figure 5.** Neurons containing classical transmitters provide heterogeneous input to insulin-producing cells (IPCs). (**A – D**) Presynaptic neurons to IPCs in the FlyWire connectome contain acetylcholine (**A**), glutamate (**B**), GABA (**C**), or unknown transmitters (**D**). Scale bar = 100 µm. (**E**) Connectivity map of presynaptic neurons containing glutamate (purple), acetylcholine (magenta), or GABA (blue) to individual IPCs (green). Connection strength is indicated with the numbers of synapses by line thickness (upper panel) and circle diameter (lower panel). Individual IPCs differ regarding their synaptic input and input strength. Note that 18 IPCs were identified in FlyWire (***Musselman et al., 2011***).

by the number of affected IPCs. For DAN, OAN, and MSN activation, the number of cells per cluster was similar in patch-clamp and calcium imaging, while for AstANs, and LKNs the fraction of IPCs per cluster differed considerably (***Figure 6B***). On a single-cell level, we found heterogeneous responses while imaging multiple cells at the same time during activation of the same ModN population. This strongly indicates that the response clusters we observed in patch-clamp recordings are not due to state-dependent shifts between recordings. Instead, the calcium imaging recordings indicated that the heterogeneous responses are a persistent feature of the IPC population. Within any given cluster, the responses appeared highly homogeneous, indicating that the modulatory input to the associated IPCs was similar within the response clusters.

To link our receptor expression analyses to the functional connectivity measured in electrophysiological recordings and calcium imaging, we analyzed the G-protein coupling scores for the receptors of interest, including multiple splice isoforms where relevant (***Sgourakis et al., 2005***) (see ***Supplementary file 1B*** for details; http://athina.biol.uoa.gr/bioinformatics/PRED-COUPLE2/). To simplify comparisons, we predicted that a neuromodulator could be excitatory if at least one of its receptors (including transcript variants) coupled to Gs or Gq proteins. Similarly, a neuromodulator was predicted to be inhibitory if at least one of its receptor variants was predicted to couple to Gi protein. When comparing the predictions and anatomical mapping results with the results from patch-clamp recordings and calcium imaging, it became apparent that the majority of ModN activation effects were correctly predicted by the receptor profile (***Figure 6C***). For example, AstA receptors were predicted

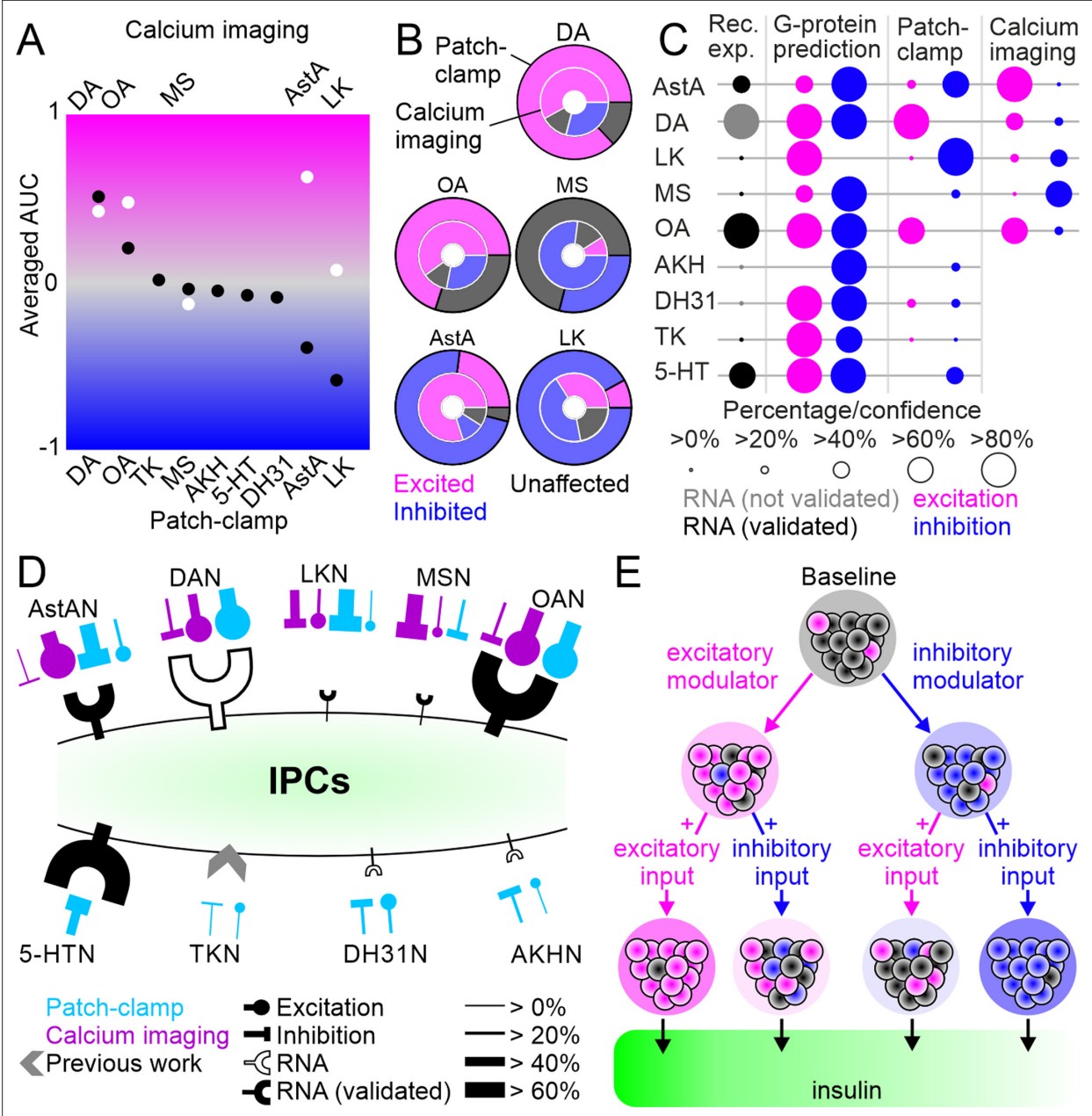

**Figure 6.** Summary of neuromodulation of insulin-producing cells (IPCs). (**A**) Comparison of overall activity shifts for all driver lines and IPCs tested between patch-clamp (black) and calcium imaging (white). (**B**) Comparison of cluster proportions between patch-clamp (outer circles) and calcium imaging (inner circles). (**C**) Comparison of, expression data, G-protein coupling prediction, patch-clamp recording, and calcium imaging. Dots indicate percentage of IPCs for expression data, patch-clamp, and calcium imaging, and the confidence for G-protein prediction. For receptor expression data, black dots indicate the expression has been validated with receptor mapping. (**D**) Simplified model of IPC modulation. Overview of receptor expression and functional connectivity in patch-clamp (blue) and calcium imaging (purple) recordings for all modulatory inputs tested. Line thickness indicates fraction of involved IPCs. Receptor expression for TK from **Birse et al., 2011**. (**E**) Simplified hypothesis of IPC population activity shifts through excitatory or inhibitory modulator input and the effects of consecutive input on insulin output.

to signal via both excitatory and inhibitory secondary messenger pathways, which could explain the heterogeneity and the differences observed between patch-clamp and calcium imaging experiments. On the other hand, we expected that LKN activation would primarily result in IPC excitation based on G-protein coupling predictions, but this contrasts with the functional connectivity as determined by calcium imaging and patch- clamp experiments. Hence, the effects of LKN activation on IPCs are likely due to indirect connections between LKNs and IPCs, and potentially other neurotransmitters

co-released by LKNs upon activation. However, for the majority of the investigated ModNs, their global effects on the IPC activity could be accurately predicted based on receptor expression and G-protein coupling, which can also account for the heterogeneity within the IPC population.

## Discussion

Our results show that the activity of IPCs in *Drosophila* is affected by multiple neuromodulatory inputs. We analyzed the receptor profile of IPCs and the effects of nine key ModN populations on the IPCs on the level of single-cell responses and population activity shifts (*Figure 6C and D*). Hence, our study provides a multi-level quantification of the complex regulation of IPCs, which underlies their ability to respond to changing metabolic demands, other internal state changes, and sensory inputs.

### General effects of ModNs on IPC activity

Neuromodulation enables multiplexing in neuronal circuits and lends flexibility to seemingly hard-wired connections. In contrast to classical, fast-acting neurotransmitters, neuromodulators can act far away from their release sites, and thus change neuronal circuit dynamics across the entire nervous system. Due to the complexity of neuromodulatory systems, studies examining their function or modes of action have often been limited to ex vivo preparations or in vitro physiological assays (*Bargmann and Marder, 2013*). Here, we leveraged the IPCs of *Drosophila* as a model system to investigate neuromodulation by combining multiple approaches, including in vivo patch-clamp recordings and calcium imaging, as well as receptor expression mapping and connectomics. Our results revealed that the IPC population activity can be shifted towards excited or inhibited states on fast timescales through the action of different ModN populations. Some ModNs have strong, immediate effects on the global IPC activity, which are suited to prime the system for pending metabolic demands by altering the excitability of IPCs and ultimately affect insulin release. Prolonged activation protocols during patch-clamp recordings and calcium imaging experiments demonstrated that these effects persist during continued activation (*Figure 2—figure supplement 2A and B*; *Figure 4—figure supplement 1A*). This indicates that the input of the respective ModNs could also affect the IPC activity on longer timescales. These effects on long timescales could in turn affect the IPCs on two levels: (1) sustained excitation could lead to a direct increase in insulin release, and (2) continuous activation of IPCs through modulatory inputs could lead to altered gene expression, e.g., resulting in increased expression of *ilps* (*Chen et al., 2016a*; *Li and Gong, 2015*). Prolonged inhibition could have the opposite effect, leading to a decrease of insulin release and ultimately decreased *ilp* expression. In both scenarios, the prolonged modulation of IPC activity and, therefore, insulin release could alter nutritional state-dependent behaviors, such as foraging (*Sudhakar et al., 2020*). Although most ModNs investigated in our study co-express other signaling molecules, e.g., additional neuropeptides and fast-acting neurotransmitters (*Croset et al., 2018*), we focus our discussion on functions of the neuromodulators that define the populations, especially in relation to insulin signaling. For example, activating DANs and OANs shifted the IPC population activity towards an excited state in patch-clamp recordings and calcium imaging. In *Drosophila*, DA is implicated in modulating locomotion, learning, and courtship behavior (*Yamamoto and Seto, 2014*; *Siju et al., 2021*; *Cohn et al., 2015*). In female flies, DA directly acts on the IPCs via DopR1, which affects ovarian dormancy via DILP signaling (*Andreatta et al., 2018*). Our results demonstrate that DANs have a fast and strong excitatory effect on IPC activity in vivo, on short time scales. However, ovarian dormancy is accompanied by diminished insulin signaling and is likely controlled over longer time scales. Hence, there might be an additional long-term effect of DA on IPCs, which we have not explored here. Our finding that OANs had strong, fast, excitatory effects on IPCs in patch-clamp recordings and calcium imaging was somewhat unexpected. This is because the actions of OA and insulin are antagonistic in several contexts. For example, OA drives an increase in locomotor activity while insulin decreases locomotion (*Bisen et al., 2024*; *Yang et al., 2015*; *Yu et al., 2016*; *Brembs et al., 2007*). Accordingly, OA is released during increased physical activity, such as flight and walking, while insulin release is inhibited (*Liessem et al., 2023*; *Suver et al., 2012*). Moreover, knockdown of the OAMB receptor in IPCs increases *Dilp2* expression (*Luo et al., 2014*; *Li et al., 2016*). However, our receptor expression analysis revealed that IPCs express a variety of OA receptors besides OAMB. In addition, OA has been reported to have varying effects in foraging behavior. While it has been demonstrated that OA, in general, mediates

hyperactivity in starved flies (*Bisen et al., 2024*; *Yang et al., 2015*), activating a specific OAN type suppresses odor tracking during foraging (*Sayin et al., 2019*). Similarly, reduced insulin levels increase odor-driven food-searching behavior (*Ko et al., 2015*; *Root et al., 2011*). Hence, OA does not always act antagonistically to insulin, which means that context-dependent differences in the effect of OA on IPCs are conceivable. Our G-protein coupling predictions indicate that the expressed OA receptors could be excitatory and inhibitory; hence, activation of OA receptors could lead to heterogeneous responses in IPCs. In addition, neurons containing OA co-express other transmitters, in particular tyramine, Glu, GABA, and ACh (*Croset et al., 2018*). Therefore, activating OANs in our experiments could lead to the release of several transmitters, which could bind to a variety of receptors resulting in an overall excitatory effect. Importantly, since we activated OANs intrinsically, these co-transmitters are likely also released under natural circumstances.

In contrast, MSN and LKN activation led to an inhibition of the IPC activity in patch-clamp and calcium imaging experiments (*Figure 6B*). MS is known to have relaxant properties on insect muscles and leads to crop enlargement in female flies during food consumption after mating (*Hadjieconomou et al., 2020*). Thus, MS enables the consumption of larger meals, which is essential for mated females to account for their higher energy demands. This could be supported by the concurrent IPC inhibition we observed, since this would result in lowered insulin levels and, therefore, a higher feeding drive (*Sudhakar et al., 2020*). Insulin signaling also contributes to the modulation of mating in flies. Reduced insulin levels reduce re-mating in mated females. Hence, the inhibitory effect of MSNs on IPCs could also act as a feedback signal to reduce premature re-mating (*Wigby et al., 2011*). In addition, AKH-producing cells also express receptors for MS (*Braco et al., 2022*). This suggests that MS could be regulating metabolic homeostasis via the modulation of DILPs and AKH in the context of reproduction, where the inhibition of IPCs is one of several parallel effects mediated by MS to modulate post-mating behaviors in females.

The overall inhibitory effect of LKNs on IPCs was more pronounced in patch-clamp recordings than in calcium imaging, which is likely partially due to the higher sensitivity of electrophysiological recordings to membrane hyperpolarization. In calcium imaging, the IPC population activity coefficient was slightly positive, while it was strongly negative in patch-clamp recordings (*Figure 6A*). However, the cluster distribution shows that the inhibited IPC clusters contained the highest numbers of IPCs during LKN activation in both methods, meaning that the majority of IPCs were inhibited (*Figure 6B*). IPCs are anatomically and functionally downstream of LKNs (*Zandawala et al., 2018*; *Yurgel et al., 2019*). For example, *Lkr* knockdown in IPCs affects starvation tolerance and sleep. Specifically, a subset of LKNs increases their activity during starvation and is required for starvation-induced suppression of sleep (*Yurgel et al., 2019*). DILPs are also implicated in the modulation of sleep, since a loss of DILPs, especially DILP2 and DILP3, reduces sleep depth and duration (*Cong et al., 2015*; *Brown et al., 2020*). Our study supports these findings by showing that LKN activation, which would occur during starvation under normal conditions, has a strong inhibitory effect on IPCs. The resulting decreased DILP release is expected to cause a reduction of sleep, mimicking the effects of starvation mediated by LKNs.

Interestingly, we observed different effects of AstAN activation when comparing patch-clamp recordings with calcium imaging. In patch-clamp recordings, AstAN activation had a strong inhibitory effect on the majority of IPCs. AstA is known to inhibit feeding and promote sleep (*Nässel and Zandawala, 2020*; *Hentze et al., 2015*; *Chen et al., 2016b*). Therefore, the inhibitory effect of AstAN activation on IPCs could lead to an immediate decrease in DILP release, which is then followed by a long-term reduction in IPC activity resulting from the AstA-driven cessation of food intake. However, we also recorded a fast and strong excitation in a small subset of IPCs during patch-clamp experiments. In addition, we observed a strong calcium influx in the majority of IPCs upon AstAN activation in calcium imaging experiments. Our G-protein coupling predictions can explain these heterogeneous effects, but also the differences between patch-clamp and calcium imaging. AstAR1 and AstAR2 are both predicted to couple to Gi proteins, suggesting an inhibitory effect of AstA. However, AstAR2, which is almost exclusively expressed in IPCs within the brain, is also predicted to couple to Gq subunits. Gq subunits activate cellular pathways that lead to an intracellular calcium release that subsequently causes calcium channels in the membrane to open and the influx of extracellular calcium (*Dhyani et al., 2020*). Therefore, the presence of AstAR2 receptors suggests the possibility of a calcium increase in parallel to the overall inhibition of the IPCs (see *Supplementary file 1B* for more

details). The heterogeneity resulting from AstAN activation in both patch-clamp and calcium imaging can, therefore, be explained by heterogeneous expression of receptors across individual IPCs, since AstAR1 and AstAR2 are not expressed in all IPCs. In addition, AstAN activation could also lead to excitatory input via co-expressed transmitters. Previous single-cell transcriptome analyses indicated that at least some AstANs co-express ACh and glutamate (*Croset et al., 2018*; *Nässel, 2018*) and our expression analyses revealed that IPCs express receptors for these transmitters (*Figure 1—figure supplement 1D*). In addition, our connectome analysis revealed that all IPCs receive input from putative cholinergic and glutamatergic neurons (*Figure 5*). Moreover, indirect pathways from the AstANs to IPCs via interneurons could also drive excitatory responses, especially considering the long latency of the excitatory effect (*Figure 3A*).

Taken together, the differences we observed for AstAN activation between patch-clamp recordings and calcium imaging emphasize the importance of utilizing parallel approaches to unravel different layers of functional effects caused by modulatory inputs to IPCs. Our receptor expression analysis complements the electrophysiological data by providing valuable information about the foundation for the heterogeneous effects we observed in the functional connectivity. The single-nucleus transcriptome analysis reveals which receptor transcripts are expressed whereas the T2A-GAL4 lines used in our anatomical analyses provide insights on which of the receptor transcripts are translated. This is based on the fact that T2A peptides induce ribosome skipping during translation. Therefore, GAL4 protein is only produced when the receptor protein is produced (*Kondo et al., 2020*; *Imura et al., 2020*). Given the lack of sensitive antibodies for GPCRs, future studies, e.g., based on CRISPR/Cas9-based protein-tagging, could provide additional insights on where receptor proteins are localized on IPCs.

## IPCs form a heterogeneous population that is differentially modulated by sets of neuromodulators acting in parallel

Since the IPCs form a cluster of morphologically indistinguishable neurons and all of them express DILPs 2, 3, and 5, they have traditionally been regarded as a homogeneous population. Recently, this view has been challenged by a study that found the mechanosensitive channel Piezo in only a subset of IPCs (*Wang et al., 2020*). We found additional proof of heterogeneity within the IPC population based on receptor expression analyses and IPC responses to ModN activation. Intriguingly, cellular and functional heterogeneity also occurs in human pancreatic β-cells. This heterogeneity is assumed to allow for dynamic transitions between states of high insulin production and cellular stress recovery (*Gutierrez et al., 2017*), which could be a general feature of insulin-releasing cells.

In our study, single-nucleus transcriptomes of IPCs and anatomical receptor mapping experiments revealed that modulatory and other receptors are not uniformly expressed across the IPC population. Instead, most receptors are only expressed in subsets of IPCs. Activating ModN populations while recording the IPC responses via patch-clamp revealed that the heterogeneous receptor profiles of individual IPCs have functional significance. Calcium imaging experiments for selected ModNs further revealed that the same modulatory input led to simultaneous, heterogeneous responses across the IPC population, within the same animal. Therefore, the heterogeneity is not based on internal state changes. Finally, analyzing connectomics data showed that IPCs also receive heterogeneous inputs via direct synapses, both regarding input profile and input strength. The question that arises from our results is whether we investigate stochastic heterogeneity, or whether the heterogeneity is a control mechanism for insulin release. Since we demonstrated similar heterogeneous receptor expression as well as similar responses to a specific modulatory input across animals, stochastic heterogeneity, as for example demonstrated in the development of photoreceptors of *Drosophila* (*Anderson et al., 2017*), is unlikely. For example, over 88% of IPCs expressed at least one receptor for dopamine, which matches our recordings in patch-clamp and calcium imaging closely, where the activation of DANs had an effect on 87% of IPCs. In addition, the G-protein coupling analysis accurately predicted a strong excitation of the IPC population accompanied by a weak inhibition by DA.

Since the heterogeneity occurs on the receptor level and in physiological responses, it is likely to be beneficial for the system. We hypothesize that heterogeneous activation of individual IPCs facilitates a more nuanced insulin release and dynamic transitions between states of high insulin demand and recovery, comparable to the heterogeneity found in human pancreatic β-cells. A fine-tuned release from the IPCs is especially crucial when considering that insulin does not only regulate

metabolic homeostasis, but also acts as a neuromodulator in the brain (*Root et al., 2011*; *Aimé et al., 2012*; *Brüning et al., 2000*). In the olfactory systems of flies and rats, insulin reduces food-searching behavior in satiated animals (*Aimé et al., 2012*) through the inhibition of a specific subset of olfactory sensory neurons (*Root et al., 2011*). Therefore, insulin release needs to be adjusted to metabolic demands, and for example, be reduced in foraging animals (*Kim et al., 2017*). We have three hypotheses for how the modulation of IPCs could alter their activity dynamics and ultimately insulin release. (1) Heterogeneous excitation and inhibition of IPCs could lead to a direct adjustment of the net output by increasing or decreasing DILP release by individual IPCs. (2) We revealed that on top of the heterogeneity, the population activity was shifted towards excitation or inhibition. Shifting the IPCs into a more excited state could increase the responsiveness of the system to enable a fast response to a secondary excitatory input (*Figure 6E*), leading to a stronger release of DILPs, for example in the context of feeding. Receiving an additional inhibitory input, on the other hand, would only lead to a moderate DILP release (*Figure 6E*). In the same fashion, shifting IPCs towards inhibition could make the system less sensitive to small changes on the input level in a state of generally low-insulin demand. This would enable a moderate insulin release upon arrival of a secondary excitatory input (*Figure 6E*). On the other hand, an additional inhibitory input would lead to a drastic shut down of IPC activity (*Figure 6E*) as observed during locomotion (*Liessem et al., 2023*). (3) The heterogeneous input could regulate the ratio of DILP2, 3, and 5 release. For example, the secretion of either DILP2 or DILP3 is selectively increased depending on the dietary conditions (*Kim and Neufeld, 2015*). Moreover, DILP3 expression, but not that of DILP2 or 5, has been tightly connected to the activity of AstANs (*Hentze et al., 2015*).

Generally, heterogeneity adds flexibility and stability to neuromodulatory systems. In the case of IPCs, this could allow a relatively small number of neurons to integrate a large variety of inputs to achieve flexible insulin release tailored to the current internal demands.

## Materials and methods

**Key resources table**

| Reagent type (species) or resource | Designation | Source or reference | Identifiers | Additional information |
|---|---|---|---|---|
| Strain (*D. melanogaster*) | UAS-NLS-mCherry | Bloomington *Drosophila* Stock Center | RRID:BDSC_38425 | Used for receptor mapping |
| Strain (*D. melanogaster*) | Oct-TyrR-T2A-GAL4 | Bloomington *Drosophila* Stock Center | RRID:BDSC_86138 | Used for receptor mapping |
| Strain (*D. melanogaster*) | 5HT7-T2A-GAL4 | Bloomington *Drosophila* Stock Center | RRID:BDSC_84592 | Used for receptor mapping |
| Strain (*D. melanogaster*) | Oct-alpha2R-T2A-GAL4 | Bloomington *Drosophila* Stock Center | RRID:BDSC_84610 | Used for receptor mapping |
| Strain (*D. melanogaster*) | AstAR1-T2A-GAL4 | *Deng et al., 2019* | | Used for receptor mapping |
| Strain (*D. melanogaster*) | MsR1-(B)-T2A-GAL4 | Bloomington *Drosophila* Stock Center | RRID:BDSC_84653 | Used for receptor mapping |
| Strain (*D. melanogaster*) | AstAR2-(A/C)-T2A-GAL4 | Bloomington *Drosophila* Stock Center | RRID:BDSC_84594 | Used for receptor mapping |
| Strain (*D. melanogaster*) | sNPFR-T2A-GAL4 | Bloomington *Drosophila* Stock Center | RRID:BDSC_84691 | Used for receptor mapping |
| Strain (*D. melanogaster*) | CCKLR17D1-T2A-GAL4 | Bloomington *Drosophila* Stock Center | RRID:BDSC_84605 | Used for receptor mapping |
| Strain (*D. melanogaster*) | CrzR-T2A-GAL4 | *Kondo et al., 2020* | | Used for receptor mapping |
| Strain (*D. melanogaster*) | Dh31R-(A/B/C)-T2A-GAL4 | Bloomington *Drosophila* Stock Center | RRID:BDSC_84625 | Used for receptor mapping |
| Strain (*D. melanogaster*) | Dop2R-T2A-GAL4 | Bloomington *Drosophila* Stock Center | RRID:BDSC_84628 | Used for receptor mapping |
| Strain (*D. melanogaster*) | LkR-GAL4 Knock-in mutant | *Zandawala et al., 2018* | | Used for receptor mapping |

*Continued on next page*

*Continued*

| Reagent type (species) or resource | Designation | Source or reference | Identifiers | Additional information |
|---|---|---|---|---|
| Strain (*D. melanogaster*) | Oct-alpha2R-Trojan-GAL4 | Bloomington *Drosophila* Stock Center | RRID:BDSC_67636 | Used for receptor mapping |
| Strain (*D. melanogaster*) | 5-HT1A-T2A-GAL4 | Bloomington *Drosophila* Stock Center | RRID:BDSC_84588 | Used for receptor mapping |
| Strain (*D. melanogaster*) | R96A08-LexA- p65-vk37::LexOp-dilp2::GFP;20x-UAS-CsChrimson- attp2/TM6b | *Liessem et al., 2023* | | Used for optogenetic activation during patch-clamp recordings |
| Strain (*D. melanogaster*) | R96A08-LexA-p65-vk37/CyO;LexOp-GCaMP6m-vk5, 20X-UAS-CsChrimson.mVenus-su(Hw) attp1/TM6 | This study | | Used for optogenetic activation during calcium imaging |
| Strain (*D. melanogaster*) | TH-GAL4 | *Friggi-Grelin et al., 2003* | | Used for optogenetic activation |
| Strain (*D. melanogaster*) | AstA-GAL4 | Bloomington *Drosophila* Stock Center | RRID:BDSC_84593 | Used for optogenetic activation |
| Strain (*D. melanogaster*) | Tdc2-GAL4 | Bloomington *Drosophila* Stock Center | RRID:BDSC_9313 | Used for optogenetic activation |
| Strain (*D. melanogaster*) | LK-GAL4 | Bloomington *Drosophila* Stock Center | RRID:BDSC_51993 | Used for optogenetic activation |
| Strain (*D. melanogaster*) | MS-GAL4 | Bloomington *Drosophila* Stock Center | RRID:BDSC_51986 | Used for optogenetic activation |
| Strain (*D. melanogaster*) | AKH-GAL4 | Bloomington *Drosophila* Stock Center | RRID:BDSC_25684 | Used for optogenetic activation |
| Strain (*D. melanogaster*) | TK-GAL4 | Bloomington *Drosophila* Stock Center | RRID:BDSC_51973 | Used for optogenetic activation |
| Strain (*D. melanogaster*) | sNPF-GAL4 | Bloomington *Drosophila* Stock Center | RRID:BDSC_51991 | Used for optogenetic activation |
| Strain (*D. melanogaster*) | DH31-GAL4 | Bloomington *Drosophila* Stock Center | RRID:BDSC_51988 | Used for optogenetic activation |
| Strain (*D. melanogaster*) | TrH-GAL4 | Yi Rao Laboratory Chinese Institute for Brain Research, Beijing (China) | CG9122 | Used for optogenetic activation |
| Strain (*D. melanogaster*) | GFP-p10 | Bloomington *Drosophila* Stock Center | RRID:BDSC_32201 | Used for patch-clamp recordings |
| Antibody | anti-RFP (primary guinea pig polyclonal) | Dr. Susan Brenner-Morton, Columbia, USA | | Enhances RFP signal, diluted 1:5000 |
| Antibody | anti-DILP2 (primary rabbit polyclonal) | *Veenstra et al., 2008* | RRID:AB_2569969 | Used to label IPCs, diluted 1:2000 |
| Antibody | anti-nc82 (primary mouse monoclonal) | DSHB | RRID:AB_2314866 | Used to label neuropils, diluted 1:500 |
| Antibody | anti-GFP (primary chicken polyclonal) | Abcam | RRID:AB_2269474 | Enhances GFP signal, diluted 1:1000 |
| Antibody | Donkey anti-rabbit Alexa Fluor 647 (secondary donkey polyclonal) | ThermoFisher Scientific | RRID:AB_2536183 | Diluted 1:1000 |
| Antibody | Goat anti-guinea pig Alexa Fluor 555 (secondary goat polyclonal) | ThermoFisher Scientific | RRID:AB_2535856 | Diluted 1:1000 |
| Antibody | Donkey anti-mouse Alexa Fluor 488 (secondary donkey polyclonal) | ThermoFisher Scientific | RRID:AB_141607 | Diluted 1:1000 |
| Antibody | Goat anti-chicken Alexa Fluor 488 (secondary goat polyclonal) | ThermoFisher Scientific | RRID:AB_2534096 | Diluted 1:200 |

*Continued on next page*

*Continued*

| Reagent type (species) or resource | Designation | Source or reference | Identifiers | Additional information |
|---|---|---|---|---|
| Antibody | Goat anti-rabbit Alexa Fluor 555 (secondary goat polyclonal) | ThermoFisher Scientific | RRID:AB_2535849 | Diluted 1:200 |
| Antibody | Goat anti-mouse Alexa Fluor 635 (secondary goat polyclonal) | ThermoFisher Scientific | RRID:AB_2536184 | Diluted 1:400 |
| Chemical compound | SigmaCote | Sigma-Aldrich | cat. no. SL2; RRID:SCR_008988 | siliconizing reagent |
| Chemical compound | Vectashield Antifade Mounting Medium | VEC-H-1000 | Biozol | - |
| Chemical compound | all-trans-retinal | Sigma-Aldrich | R2500 | Used in optogenetic experiments |
| Software, algorithm | OCULAR | OCULAR | | Image acquisition software (patch clamp) |
| Software, algorithm | Micro-Manager 2.0 for Image J | *Edelstein et al., 2014* | | Image acquisition software (calcium imaging) |
| Software, algorithm | pCLAMP 10 | Molecular Devices | | Used for recordings (Optogenetic activation and patch clamp) |
| Software, algorithm | MATLAB R2021a | The Mathworks | | Used for data analysis and statistical testing |
| Software, algorithm | Jupyter Notebook | Project Jupyter | | Pre-processing of calcium imaging data |

BDSC = Bloomington *Drosophila* Stock Center.

## Fly husbandry

All flies were reared on standard fly food (cornmeal-agar-molasses medium) and kept in a climate chamber at 60% relative humidity, 25 °C, and a 12 hr:12 hr light/dark cycle. Experiments were performed on mated females 3–6 d after eclosion. For experiments with optogenetic activation of CsChrimson, 300 µM all-trans-retinal (R2500, Sigma-Aldrich, Steinheim, Germany) was added to standard fly food and experimental flies were kept on that food after eclosion in darkness until usage.

## Fly genotypes

The key resources table shows all genotypes used for anatomical receptor mapping and activation experiments as well as brain and VNC anatomical labeling.

## Immunohistochemistry

### Anatomical receptor mapping

Flies of the F1 generation from crosses with GAL4 lines from the key resource table and UAS-NLS-mCherry were used for anatomical receptor mapping. We primarily used the recently generated receptor-T2A-GAL4 knock-in lines *Deng et al., 2019*; *Kondo et al., 2020* to drive expression of nuclear mCherry. Since the T2A-GAL4 knock-in lines were generated by inserting the T2A-GAL4 coding sequence right before the stop codon for the gene, an accurate expression can be assumed since the GAL4 is expressed in the same temporal and spatial patterns, and at similar levels, as the native receptor. Flies were prefixed at room temperature for 2.5 hr in a 2 mL tube with 1.5 mL 4% paraformaldehyde and 20 µL 0.5% PBT (phosphate-buffered saline with 0.5% TritonX-100; Sigma-Aldrich, Steinheim, Germany). The fixed flies were washed 4×15 min each with 0.5% PBT. Brains were dissected afterwards and collected directly in 400 µL blocking solution (5% NGS in PBT) and incubated for 1.5 hr to block non-specific binding sites. Next, samples were incubated in primary antibodies (key resources table) diluted in blocking solution for 48 hr at 4 °C. Afterwards, the brains were washed 4×15 min each in PBST at room temperature before they were incubated in the secondary antibodies (key resources table) for 48 hr at 4 °C. Next, the samples were washed for 4×15 min in PBT, followed by two final washing steps in PBS for 15 min each before they were mounted in Fluoromount-G (Invitrogen, Waltham, MA, USA). Z-stacks were acquired using a Leica TCS SPE confocal microscope

(Leica Microsystems, Wetzlar, Germany) equipped with Leica Microsystems LAS AF software, a photo-multiplier tube, and 488, 532, and 635 nm solid-state lasers for excitation. We used a 20x glycerol immersion objective for whole-mount scans and 40×oil immersion objective for detailed images. All images were captured at a resolution of 1024×1024 pixels. To reduce background noise, all frames were captured two to three times and then averaged. Images were analyzed with ImageJ.

## Expression analysis of driver lines used for activation experiments

To extract the brain and VNC from flies of the driver lines we used in the optogenetic activation experiments, flies were anesthetized on ice and then fixed in 4% PFA in 0.1 M PBS for 3 hr followed by three subsequent washes in 0.1 M PBS. They were then dissected in 0.1 M PBS at room temperature. Prior to dissection, each fly was shortly submerged in 70% ethanol for ~30 s. The dissected brains were then washed in PBT for 3×15 min. To block non-specific binding sites the brains and VNCs were pre-incubated in 10% Normal Goat Serum (NGS) in PBT (blocking buffer) overnight at 4 °C. The primary antibodies (key resources table) were diluted in blocking buffer and this solution was then used to incubate brains and VNCs for 48 hr at 4 °C. Next, the tissues were washed 3×15 min. in PBT and, then incubated overnight at 4 °C (or for 3 hr at RT) with the secondary antibodies diluted in blocking buffer (key resources table). After washing for 3×15 min. with PBT, the solution was replaced with Vectashield Antifade Mounting Medium (H-1000, Vector Laboratories, CA, USA) and the brains and VNCs were mounted on adhesion slides (Polysine slides, Thermo Scientific, Waltham, MA, USA). After covering the samples with Vectashield under a coverslip, the edges were sealed with nail polish and stored at 4 °C until images were taken. Fluorescence images were acquired with a confocal laser scanning microscope (CLSM, Leica TCS SP8 WLL) via the Leica Application Suite X (LAS X, Leica Microsystems) using HC PL APO 10x/0.4, HC PL APO 20×/0.75 IMM objectives. Fluorescence signals were detected in serial stacks with a resolution of 1024×1024 pixels using three channels in sequential scanning mode. The GFP signal, which labeled the respective MN population of the GAL4 lines in the key resources table was acquired using an excitation laser wavelength of 488 nm and a HyD detector tuned to 498–540 nm. For Alexa Fluor 555 visualization, which was used to label the IPCs via the DILP2 antibody staining, we excited the fluorophore with the laser set to 555 nm and detected the emission with a PMT tuned to 565–620 nm. In the third channel, Alexa Fluor 635 bound to the nc82 antibody was excited with a wavelength of 633 nm and the emitted light was detected by a second PMT tuned to 643–693 nm. The image stacks were saved as a lif file from which maximum projections and scale bars were created using the Leica Application Suite X. Images were analyzed with ImageJ.

## In vivo whole-cell patch-clamp recordings

Fly handling and preparation were performed as described in *Ache et al., 2019*. All experiments were performed at room temperature under daylight conditions. Before preparation, flies were cold anesthetized at 4 °C for immobilization and the head and thorax were positioned in a custom-made shim plate fly holder and mounted with UV glue (Proformic C1001, VIKO UG, Munich, Germany). After removing the forelegs, the remaining stumps and the proboscis were glued together for maximal stability. Afterwards, a window was cut into the posterior cuticle of the head. Trachea, fat tissue, and the ocellar ganglion were removed to expose the brain and gain access to the IPC cell bodies in the pars intercerebralis (PI). The fly was then transferred with the fly holder into a fixed stage fluorescence upright microscope (Olympus BX51WIF, Olympus Corporation, Japan) or a customized Slicescope (Scientifica, Uckfield, UK). Live images of the fly brain were acquired with a high-resolution camera (SciCam Pro, Scientifica, US) using an image acquisition software (OCULAR, Digital Optics Limited, Auckland, NZ). The brain was continuously perfused with carbonated (95% $O_2$ and 5% $CO_2$) extracellular saline containing 103 mM NaCl, 3 mM KCl, 5 mM N-[Tris(hydroxymethyl)methyl]–2-aminoethanesulfonic acid, 10 mM D-(+)-glucose, 8 mM D-(+)-Trehalose dihydrate, 26 mM $NaHCO_3$, 1 mM $NaH_2PO_4$, 1.5 mM $CaCl_2.2H_2O$, 4 mM $MgCl_2.6H_2O$, and osmolarity adjusted to 273–275 mOsm (*Gouwens and Wilson, 2009*) during the preparation and the experiment. For enzymatic dissociation of the neural sheath from the brain above the IPCs, 0.025% (w/v in extracellular saline) collagenase (Sigma-Aldrich #C5138) was applied locally in the PI region using a blunt thin-walled pipette. After identifying the GFP-expressing IPC cell bodies, whole-cell patch-clamp recordings were performed using thick-walled patch pipettes (4–8 MΩ resistance) containing intracellular saline (140 mM potassium aspartate, 10 mM HEPES, 1 mM EGTA, 4 mM MgATP, 0.5 mM $Na_3$ GTP, and 1 mM KCl, adjusted

to 260–275 mOsm, pH 7.3). Continuous time series of the membrane potential and spike events were captured in current clamp mode with an AxoPatch MultiClamp 200B (Molecular Devices LLC., San Jose, CA, USA). A part of data were also collected with a MultiClamp 700B amplifier (Molecular Devices). All data were recorded with a Digidata 1440 A analog-digital converter (Molecular Devices) controlled by the pCLAMP 10 software using a 10 kHz low-pass filter and a 20 kHz sampling rate. All optogenetics experiments were carried out using a 625 nm LED adjusted to an intensity of ~4.4 mW/cm$^2$ at the fly for CsChrimson activation. The LED was operated using TTL triggers controlled by preset protocols in pCLAMP. Each recording consisted of 10 activations with a stimulation length of 100 ms and a 10 s inter-stimulus-intervals. The TTL triggers were recorded simultaneously with the membrane potential and stored in abf format (Axon Binary File).

### In vivo calcium imaging

Calcium imaging experiments were preceded by the same preparation protocol as for whole-cell patch-clamp recordings. The calcium concentration of the extracellular saline solution was increased to match the saline used for calcium imaging used before (*Seelig and Jayaraman, 2015*), with a final concentration of 2.5 mM CaCl$_2$.2H$_2$O All calcium imaging data were acquired with a Prime BSI Express Scientific CMOS camera (Teledyne Photometrics, Tucson, AZ, USA) in an in vivo fluorescence microscope (SliceScope, Scientifica) controlled by the ImageJ based software Micro-Manager 2.0 (*Edelstein et al., 2014*). Time series were recorded in a single z-plane at a framerate of 10 Hz for 2 min per trial and stored as tiff stacks. The x and y dimensions of the image frames were adjusted individually to the respective position of the IPCs in each fly. The genetically encoded calcium indicator GCaMP6m (*Chen et al., 2013*) was expressed in IPCs, while CsChrimson was expressed in the respective MN driver line (key resources table). GCaMP6m was then excited via a tunable LED (pE-4000 universal fluorescence illumination system, CoolLED Ltd., Andover, UK) set to 470 nm. The light beam was directed through a GFP filter set (Chroma, Bellow Falls, VT, USA), including a 495 nm long-pass filter and an excitation filter with a bandwidth of 450–490 nm. An emission filter with a bandwidth of 500–550 nm blocked unspecific background light before emitted light from the specimen was detected by the sCMOS sensor of the camera. CsChrimson was activated in the same manner as in patch-clamp recordings, but the activation protocols were adapted to the 2 min trials. The LED trigger signal was stored together with a trace of TTL pulses for each frame of the camera as abf files. We used different lengths of activation: 5×100 ms, 5×1 s, 2×5 s, or 1×15 s per time series, with the 5 s protocol being used for the majority of the analysis and figures.

## Data analysis

### Single-nucleus transcriptomic analysis of IPCs

The pre-processed single-nucleus transcriptome data for IPCs was downloaded from https://www.flycellatlas.org/ *Li et al., 2022*. This data was further processed and analyzed with the Seurat package (version 4.1.1; https://satijalab.org/seurat/articles/pbmc3k_tutorial.html) in R-Studio (*Hao et al., 2021*). The original data comprised of 658 single-cell transcriptomes, derived from 250 males and 250 females, and sequenced using the Smart-seq2 technology. We used a stringent criterion (*Ilp2* >0 & *Ilp3* >0 & *Ilp5* >0) to retrieve 392 IPC transcriptomes (from 194 male and 198 female flies transcriptomes) which express all three *Ilps*. To determine if a transcript is present in the IPC transcriptomes, we used a 5% cutoff to reduce false positives. This cutoff is equivalent to expression in 0.8 IPCs out of 16 on average in an individual fly, and hence less than one IPC in the entire population. Since we used 392 IPC transcriptomes in our analysis, this cutoff means that expression in less than 20 IPCs will be deemed false positive. Standard commands in Seurat were used to perform the t-distributed Stochastic Neighbor Embedding (t-SNE) analysis and create dot plots and feature plots. Heatmaps were generated using the heatmap package (1.0.12).

### Patch-clamp

For patch-clamp recordings, all analysis steps and figure creations were carried out using custom-written MATLAB scripts (Mathworks, Natick, USA). To analyze the recorded patch-clamp traces we first detected all spike events using a custom-written function. To remove high-frequency noise from the traces, a median filter was applied using a time window of 250ms to each of the 10 time series that compose an optogenetic stimulation experiment. We centered all values around zero by subtracting

the median of each trace and we calculated the median of all 10 recordings to estimate the global response of each cell across trials. In order to provide a measure of the baseline activity variability, the 10th and 90[th] percentiles of all 10 filtered traces were computed in a time window of 900 ms starting 100 ms after recording onset. Thus, we established two thresholds to evaluate the effects of MN populations on the IPC activity. The median of the 10 traces was then used to represent the global response of the recorded IPC, and for the clustering into excited, inhibited, and non-responsive neurons. If the median trace was above the 90[th] percentile in a time window of 1 s starting at activation onset for longer than 50 consecutive milliseconds, the cell was classified as 'excited.' Conversely, an IPC was classified as 'inhibited' if its activity was below the 10th percentile for 50 consecutive milliseconds, and as 'unaffected' if neither of the above cases applied. We ran this analysis across all recorded IPCs and across all tested activation experiments. Thus, we quantified the percentage of IPCs that were excited, inhibited, or unaffected by the activation of each MN population. To further analyze the spiking activity, we computed the firing rate of each IPC across the 10 trials using non-overlapping time bins of 250 ms. To obtain an overall representation of the population response across all cells, we computed the mean of the median traces of all recorded cells and the mean of their individual firing rates, providing an overview of the collective neuronal activity during the activation experiments. The latency analysis of postsynaptic potentials was conducted using a Savitzky-Golay filter of fourth-order polynomial with four distinct time windows (400 ms, 200 ms, 100 ms, 50 ms) to capture fast dynamics of IPC activity while filtering out individual action potentials. After individual trial traces were filtered, we subtracted the median of each time series to center them to zero. We calculated the median of all 10 recordings and determined the 10[th] and 90[th] percentiles of baseline activity, following the previously described procedure for each time window. For cells previously classified as 'excited' or 'inhibited' we determined the latency of the stimulus as the time delay, following stimulus onset, at which the median of all 10 trials crossed the superior threshold (for excitation) or the inferior threshold (for inhibition). To estimate the overall latency, we averaged the values obtained from the four filter window lengths for each cell. Finally, to compare the magnitude of the IPC modulation via different modulatory populations, we computed the area under the curve (AUC) of the median traces with a trapezoidal numerical integration from activation onset to 1 s after onset. In the same manner, we calculated the baseline AUC from 1 s before stimulus onset to onset and then compared both values, p-values were calculated using the Wilcoxon signed-rank test. To compute a simple effect coefficient, we averaged the AUCs across all IPCs for the respective driver line, which indicated the activation shift of the population activity towards excitation or inhibition. By normalizing them to 1, with 1 being the maximum AUC value across all excited cells of the respective driver line, we were then able to compare the effects between activation experiments.

## Calcium imaging

The pre-processing of all calcium imaging data was carried out with custom-written Python code based on the fly2p module as described before (*Liessem et al., 2023*). To pre-process the imaging data, each frame of a trial was spatially filtered with a two-pixel-wide Gaussian filter and x/y motion corrected by using phase correlation to a reference image that was generated from the mean over 50 frames in midst of a trial. Fluorescence changes were computed as $\Delta F/F = (F-F0)/F0$ for each pixel over all frames, with F0 being defined as fluorescence baseline averaged over the 10% of lowest-intensity frames in each trial. Next, individual IPC cell bodies were manually delimited as regions of interest (ROIs). From these ROIs, mean $\Delta F/F$ values were calculated throughout the time series by averaging pixels within the ROI, and stored as csv files. All further analysis steps were conducted using custom-written MATLAB scripts. First, the $\Delta F/F$ traces were aligned and visualized with the respective LED trigger signal. The distribution of the $\Delta F/F$ values of all ROIs was calculated in a 5 s window before the first stimulus across all four trials for each animal as a baseline. The $\Delta F/F$ traces for each ROI from 5 s before stimulus onset to 5 s after stimulus cessation were then superimposed and the mean of these traces was quantified the percent of IPCs per cluster based on the total number of IPCs recorded in every animal. To quantify how the activation of the respective modulatory neurons shifted the population activity of the IPCs, we also calculated an effect coefficient. To this end, we first subtracted the median of a 1 s baseline window before the stimulus onset from the $\Delta F/F$ traces and then calculated the AUC as described for patch-clamp experiments for each ROI, from activation onset to 5 s after activation offset. Finally, we compared the AUC values of the traces without baseline

**Table 1.** p-values for calcium imaging area under the curves (AUCs) before and after activation.

| Line | p-values Global | Excited | Unaffected | inhibited |
|---|---|---|---|---|
| AstAN | <0.0001 | <0.0001 | 0.6875 | 0.0078 |
| OAN | <0.0001 | <0.0001 | 0.9101 | <0.0001 |
| DAN | 0.0036 | <0.0001 | 0.0195 | 0.0001 |
| LKN | 0.3684 | <0.0001 | 0.0495 | <0.0001 |
| MSN | <0.0001 | 0.1562 | 0.3593 | <0.0001 |

subtraction in a 5 s time window before activation onset to the AUC values in a time window spanning from activation onset to 5 s after activation ceased. These two values were statistically compared, p-values were calculated using a Wilcoxon signed-rank test (*Table 1*). The mean ΔF/F traces for each IPC across all animals tested in the respective driver line were then visualized in a heatmap, normalized to the highest ΔF/F as 1 for each animal. To analyze the effect of each MN population on the IPC activity further, we employed a threshold-based clustering analysis. To this end, we calculated the individual 20th and 80th percentiles of the ΔF/F value distributions in the baseline window for each animal and each IPC separately and used them as upper and lower thresholds. If the ΔF/F trace rose above the upper threshold for a continuous frame number equal 25% of all frames in a window after stimulus onset until 5 s after stimulus offset, this IPC was classified as 'excited.' If the trace decreased beneath the lower threshold for the same time window, the IPC was classified as 'inhibited.' If an increase or decrease did not meet the time criteria or if the trace stayed between the thresholds, the cell was classified as 'unaffected.' Afterwards, the mean and standard deviation were calculated for each cluster and each tested driver line. Then, we quantified the percent of IPCs per cluster based on the total number of IPCs recorded in every animal. To quantify how the activation of the respective modulatory neurons shifted the population activity of the IPCs, we also calculated an effect coefficient. To this end, we first subtracted the median of a 1 s baseline window before the stimulus onset from the ΔF/F traces and then calculated the AUC as described for patch-clamp experiments for each ROI, from activation onset to 5 s after activation offset. Finally, we compared the AUC values of the traces without baseline subtraction in a 5 s time window before activation onset to the AUC values in a time window spanning from activation onset to 5 s after activation ceased. These two values were statistically compared, p-values were calculated using a Wilcoxon signed-rank test.

## FlyWire whole-brain connectome

To analyze synaptic connections between IPCs and presynaptic partners, we used the FlyWire whole-brain connectome (*Dorkenwald et al., 2022*), which is based on the full adult fly brain (FAFB) (*Zheng et al., 2018*). All IPCs (*Reinhard et al., 2023*) and their presynaptic partners (synapse threshold ≥5) were identified by using the Connectome Data explorer (Codex) (*Lin et al., 2024*). We then exported the pre- and postsynaptic neuron IDs, as well as predicted transmitters in the presynaptic neurons. For the transmitter prediction score, we used a stringent threshold of 0.62 (*Eckstein et al., 2023*), for all predictions below that threshold the neurotransmitter type was labeled as 'unknown'. The presynaptic inputs were then pooled based on the predicted transmitters and visualized as a connectivity map with the individual IPCs and synaptic weights using a custom-written MATLAB Script. All figures were finalized with CorelDRAW (Cascade Parent Limited, Ottawa, Canada).

## Acknowledgements

We thank Haluk Lacin (University of Missouri-Kansas City) for providing flylines for activation experiments, Jan A Veenstra (University of Bordeaux) for providing the DILP-2 antibody, and Susan Morton for providing the RFP antibody. We are grateful to Konrad Öchsner for technical assistance and to Charlotte Helfrich-Förster and Wolfgang Rössler (all Julius-Maximilians-Universität of Würzburg) for sharing resources; We also thank Nicole Enslinger for preliminary calcium imaging experiments and Emilia Derksen (both JMU) for technical support. We are grateful to Dick R Nässel (Stockholm University), Tanja A Godenschwege (Florida Atlantic University), Mattias Alenius (Umeå University), and Chris

J Dallmann (JMU) for valuable feedback on the manuscript. We thank the Princeton FlyWire team and members of the Murthy and Seung labs, as well as members of the Allen Institute for Brain Science, for development and maintenance of FlyWire (supported by BRAIN Initiative grants MH117815 and NS126935 to Murthy and Seung). We also acknowledge members of the Princeton FlyWire team and the FlyWire consortium for neuron proofreading and annotation.This work was supported by a grant from the Deutsche Forschungsgemeinschaft (DFG) to JMA via the Emmy Noether program (DFG AC 371/1–1), by a grant from the DFG to JMA as part of the NSF/CIHR/DFG/FRQ/UKRI-MRC Next Generation Networks for Neuroscience (Neuronex) Program (DFG AC 371/2–1), by a grant to JMA from the Bavaria California Technology Center (BaCaTec), and by a grant from the DFG to MZ (DFG ZA1296/1-1).

## Additional information

### Competing interests

Meet Zandawala: Reviewing editor, eLife. The other authors declare that no competing interests exist.

### Funding

| Funder | Grant reference number | Author |
| --- | --- | --- |
| Deutsche Forschungsgemeinschaft | DFG AC 371/1-1 | Jan M Ache |
| Deutsche Forschungsgemeinschaft | DFG AC 371/2-1 | Jan M Ache |
| Deutsche Forschungsgemeinschaft | DFG ZA1296/1-1 | Meet Zandawala |

The funders had no role in study design, data collection and interpretation, or the decision to submit the work for publication.

### Author contributions

Martina Held, Conceptualization, Data curation, Software, Formal analysis, Validation, Investigation, Visualization, Methodology, Writing – original draft, Writing – review and editing; Rituja S Bisen, Conceptualization, Data curation, Formal analysis, Validation, Investigation, Visualization, Methodology, Writing – review and editing; Meet Zandawala, Conceptualization, Resources, Data curation, Software, Formal analysis, Supervision, Funding acquisition, Validation, Investigation, Visualization, Methodology, Writing – original draft, Writing – review and editing; Alexander S Chockley, Data curation, Investigation, Writing – review and editing; Isabella S Balles, Data curation, Formal analysis, Validation, Investigation, Visualization; Selina Hilpert, Data curation, Validation, Investigation, Visualization; Sander Liessem, Data curation, Validation, Investigation, Writing – review and editing; Federico Cascino-Milani, Data curation, Software, Formal analysis, Validation, Investigation, Visualization, Methodology, Writing – review and editing; Jan M Ache, Conceptualization, Resources, Software, Formal analysis, Supervision, Funding acquisition, Validation, Visualization, Methodology, Writing – original draft, Project administration, Writing – review and editing

### Author ORCIDs

Martina Held ⓘ https://orcid.org/0000-0001-5360-8333
Rituja S Bisen ⓘ https://orcid.org/0000-0002-8312-7191
Meet Zandawala ⓘ https://orcid.org/0000-0001-6498-2208
Jan M Ache ⓘ https://orcid.org/0000-0001-7355-7860

Reviewer #1 (Public review): https://doi.org/10.7554/eLife.99548.3.sa1
Reviewer #2 (Public review): https://doi.org/10.7554/eLife.99548.3.sa2
Author response https://doi.org/10.7554/eLife.99548.3.sa3

## Additional files

### Supplementary files
Supplementary file 1. Tables listing abbreviations used for receptors, neuropeptides, and biogenic amines (**A**) and G-protein prediction scores for selected receptors in insulin-producing cells (IPCs) (**B**).

MDAR checklist

### Data availability
The raw data contained in all figures can be accessed under the https://doi.org/10.6084/m9.figshare.27276828.v1.

The following dataset was generated:

| Author(s) | Year | Dataset title | Dataset URL | Database and Identifier |
|---|---|---|---|---|
| Held M, Bisen RS, Zandawala M, Chockley AS, Balles IS, Hilpert S, Liessem S, Cascino-Milani F, Ache JM | 2025 | Data for: Aminergic and peptidergic modulation of Insulin-Producing Cells in *Drosophila* | https://doi.org/10.6084/m9.figshare.27276828 | figshare, 10.6084/m9.figshare.27276828 |

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
