## [Editor Report · eLife Assessment]

This **fundamental** study comprehensively characterizes insulin producing cells (IPCs) resident in the *Drosophila melanogaster* brain. A **compelling** experimental tour de force, the combination of connectomics, mapping of receptors for neuromodulators, electrophysiological recordings, calcium imaging and optogenetics demonstrates that IPCs operate as a functionally heterogeneous population, as necessary to address continuously changing metabolic demands. These findings will be of interest to both neuroscientists and physiologists seeking to study context-dependent neuroendocrine regulation.

---

## [Referee Report · Reviewer #1 (Public review)]

Summary:

Insulin is crucial for maintaining metabolic homeostasis, and its release is regulated by various pathways, including blood glucose levels and neuromodulatory systems. The authors investigated the role of neuromodulators in regulating the dynamics of the adult *Drosophila* IPC population. They showed that IPCs express various receptors for monoaminergic and peptidergic neuromodulators, as well as synaptic neurotransmitters with highly heterogeneous profiles across the IPC population. Activating specific modulatory inputs, e.g. dopaminergic, octopaminergic or peptidergic (Leucokinin) using an optogenetic approach coupled with in vivo electrophysiology unveiled heterogeneous responses of individual IPCs resulting in excitatory, inhibitory or no responses. Interestingly, calcium imaging of the entire IPC population with or without simultaneous electrophysiological recording of individual cells showed highly specific and stable responses of individual IPCs suggesting their intrinsic properties are determined by the expressed receptor repertoire. Using the adult fly connectome they further corroborate the synaptic input of excitatory and inhibitory neuronal subsets of IPCs. The authors conclude that the heterogeneous modulation of individual IPC activity is more likely to allow for flexible control of insulin release to adapt to changes in metabolic demand and environmental cues.

Strengths:

This study provides a comprehensive, multi-level analysis of IPC properties utilizing single-nucleus RNA sequencing, anatomical receptor expression mapping, connectomics, electrophysiological recordings, calcium-imaging and an optogenetics-based 'intrinsic pharmacology' approach. It highlights the heterogeneous receptor profiles of IPCs, demonstrating complex and differential modulation within the IPC population. The authors convincingly showed that different neuromodulatory inputs exhibit varied effects on IPC activity and simultaneous occurrence of heterogeneous responses in IPCs with some populations exciting a subset of IPCs while inhibiting others, showcasing the intricate nature of IPC modulation and diverse roles of IPC subgroups. The temporal dynamic of IPC modulation showed that polysynaptic and neuromodulatory connections play a major role in IPC response. The authors demonstrated that certain neuromodulatory inputs, e.g. dopamine, can shift the overall IPC population activity towards either an excited or inhibited state. The study thus provides a fundamental entry point to understanding the complex influence of neuromodulatory inputs on the insulinergic system of *Drosophila*.

Weakness:

GPCRs are typically expressed at low levels and while the transcriptomic and reporter expression analysis by the authors was comprehensive, challenges remain to fully validate receptor expression and function. It will thus require future studies to elucidate how these modulatory inputs affect insulin release and transcriptional long-term changes using receptor-specific manipulation and readouts for insulin release. Similarly, optogenetically driven excitation of modulatory neuronal subsets limits the interpretation of the results due to the possibly confounding direct or indirect effect of fast synaptic transmission on IPC excitation/inhibition, and the broad expression of some neuromodulatory lines used in this analysis.

Despite these limitations that are beyond the scope of this study, the conclusions made by the authors are balanced and well supported by the data provided. Moreover, their detailed and thorough analysis of IPC modulation will have a significant impact on the field of metabolic regulation to understand the complex regulatory mechanism of insulin release, which can now be studied further to provide insight about metabolic homeostasis and neural control of metabolic processes.

---

## [Referee Report · Reviewer #2 (Public review)]

Summary:

Held et al. investigated the distinct activities of Insulin-Producing Cells (IPCs) by electrophysiological recordings and calcium imaging. In the brain of the fruit fly *Drosophila melanogaster*, there are approximately 16 IPCs that are analogous to mammalian pancreatic beta cells and provide a good model system for monitoring their activities in vivo. The authors performed single-nucleus RNA sequencing analysis to examine what types of neuromodulatory inputs are received by IPCs. A variety of neuromodulatory receptors are expressed heterogeneously in IPCs, which would explain the distinct activities of IPCs in response to the activations of neuromodulatory neurons. The authors also conducted the connectome analysis and G-protein prediction analysis to strengthen their hypothesis that the heterogeneity of IPCs may underlie the flexible insulin release in response to various environmental conditions.

Strengths:

The authors succeeded patch-clamp recordings and calcium imaging of individual IPCs in living animals at a single-cell resolution, which allows them to show the heterogeneity of IPCs precisely. They measured IPC activities in response to 9 types of neurons in patch-clamp recordings and 5 types of neurons in calcium imaging, comparing the similarities and differences in activities between two methods. These results support the idea that the neuromodulatory system affects individual IPC activities differently in a receptor-dependent manner. This work explores the fundamental properties of IPCs that may contribute to the neuroendocrine regulation of insulin-like peptides in maintaining metabolic homeostasis.

Weaknesses:

It remains unknown how much extent the heterogeneity of IPC activities in a short time scale is relevant to the net output, a release of insulin-like peptides in response to metabolic demands in a relatively longer time scale. The authors can test their hypothesis by manipulating the heterogenous expressions of receptor genes in IPCs and examine IPC activities in the future. Moreover, while the authors focus on IPC activities, they did not show the activation of the neuromodulatory inputs and the net output of insulin levels in the data. The readers might want to know which neurons are indeed activated to send signals to IPCs and how IPC activities result in the secretion of insulin peptides.

---

## [Author Response]

The following is the authors’ response to the original reviews.

**Reviewer #1 (Public review):**
Summary:Insulin is crucial for maintaining metabolic homeostasis, and its release is regulated by various pathways, including blood glucose levels and neuromodulatory systems. The authors investigated the role of neuromodulators in regulating the dynamics of the adult *Drosophila* IPC population. They showed that IPCs express various receptors for monoaminergic and peptidergic neuromodulators, as well as synaptic neurotransmitters with highly heterogeneous profiles across the IPC population. Activating specific modulatory inputs, e.g. dopaminergic, octopaminergic or peptidergic (Leucokinin) using an optogenetic approach coupled with in vivo electrophysiology unveiled heterogeneous responses of individual IPCs resulting in excitatory, inhibitory or no responses. Interestingly, calcium imaging of the entire IPC population with or without simultaneous electrophysiological recording of individual cells showed highly specific and stable responses of individual IPCs suggesting their intrinsic properties are determined by the expressed receptor repertoire. Using the adult fly connectome they further corroborate the synaptic input of excitatory and inhibitory neuronal subsets of IPCs. The authors conclude that the heterogeneous modulation of individual IPC activity is more likely to allow for flexible control of insulin release to adapt to changes in metabolic demand and environmental cues.Strengths:This study provides a comprehensive, multi-level analysis of IPC properties utilizing single-nucleus RNA sequencing, anatomical receptor expression mapping, connectomics, electrophysiological recordings, calcium-imaging and an optogeneticsbased 'intrinsic pharmacology' approach. It highlights the heterogeneous receptor profiles of IPCs, demonstrating complex and differential modulation within the IPC population. The authors convincingly showed that different neuromodulatory inputs exhibit varied effects on IPC activity and simultaneous occurrence of heterogeneous responses in IPCs with some populations exciting a subset of IPCs while inhibiting others, showcasing the intricate nature of IPC modulation and diverse roles of IPC subgroups. The temporal dynamic of IPC modulation showed that polysynaptic and neuromodulatory connections play a major role in IPC response. The authors demonstrated that certain neuromodulatory inputs, e.g. dopamine, can shift the overall IPC population activity towards either an excited or inhibited state. The study thus provides a fundamental entry point to understanding the complex influence of neuromodulatory inputs on the insulinergic system of *Drosophila*.

We thank the reviewer for endorsing our study as a fundamental entry point to understanding the complex neuromodulation of the insulin system.

Weakness:GPCRs are typically expressed at low levels and while the transcriptomic and reporter expression analysis was comprehensive, both approaches have the caveat that they do not allow validating protein level expression. Thus, some receptors might have been missed while others might be false positives. The authors acknowledged the challenges in accurately accessing receptor expression in complex modulatory systems indicating there are limitations in full understanding of the receptor profiles of IPCs.

We agree with the reviewer and acknowledge that both the transcript and protein expression need to be examined in order to obtain higher confidence in receptor expression profiles. The T2A-GAL4 lines used in our anatomical analyses do in fact provide insights into which of the receptor transcripts are translated. We added the following statement to the discussion section to clarify this approach “The singlenucleus transcriptome analysis reveals which receptor transcripts are expressed whereas the T2A-GAL4 lines used in our anatomical analyses provide insights on which of the receptor transcripts are translated. This is based on the fact that T2A peptides induce ribosome skipping during translation. Therefore, GAL4 protein is only produced when the receptor protein is produced(42,88).”

While this study provides valuable insights into the heterogeneity of IPC responses and receptor expression, it will require future studies to elucidate how these modulatory inputs affect insulin release and transcriptional long-term changes. The authors further analyzed male and female snRNAseq data and claimed that the differences in receptor expression were minimal. The experimental analyses used mated females only and while the study is very complete in this respect, it would have been extremely interesting to compare male flies in terms of their response profiles.

We thank the reviewer for acknowledging that long-term effects on release and transcript levels go beyond the scope of this study and agree that these questions should be addressed in future investigations. Concerning the differences between females and males: we did not find significant differences in the snRNAseq data between the two sexes. Moreover, a parallel study from our lab found no differences between males and females in IPC baseline activity (Bisen et al. 2024, eLife https://doi.org/10.7554/eLife.98514.1). We therefore did not follow this path for the present study. We explained our reasoning in the results section of our paper, by adding: “Since there were little differences in receptor expression between males and females (Fig. S1C), we used the transcriptomes from both sexes for all subsequent analyses.” in the transcriptome section, and “Since baseline recordings from IPCs, in addition to our transcriptomic analysis, revealed no significant difference between male and female flies(26), we only used mated females for our physiological experiments.” in the transition to the physiology section of our manuscript.

Lastly as also pointed out by the authors, their approach of using optogenetically driven excitation of modulatory neuronal subsets limits the interpretation of the results due to the possibly confounding direct or indirect effect of fast synaptic transmission on IPC excitation/inhibition, and the broad expression of some neuromodulatory lines used in this analysis.

We agree that our results are limited to general effects of neuronal populations rather than individual neurons or specific inputs, and that it is generally hard to untangle effects of fast transmitters from those of modulatory inputs. However, we believe that we are careful in presenting and interpreting our results in this regard.

Overall, however, the conclusions of this study are well supported by the data provided by the authors. Moreover, their detailed and thorough analysis of IPC modulation will have a significant impact on the field of metabolic regulation to understand the complex regulatory mechanism of insulin release, which can now be studied further to provide insight about metabolic homeostasis and neural control of metabolic processes.

We thank the referee kindly for these comments!

**Reviewer #2 (Public review):**
Summary:Held et al. investigated the distinct activities of Insulin-Producing Cells (IPCs) by electrophysiological recordings and calcium imaging. In the brain of the fruit fly *Drosophila melanogaster*, there are approximately 14 IPCs that are analogous to mammalian pancreatic beta cells and provide a good model system for monitoring their activities in vivo. The authors performed single-nucleus RNA sequencing analysis to examine what types of neuromodulatory inputs are received by IPCs. A variety of neuromodulatory receptors are expressed heterogeneously in IPCs, which would explain the distinct activities of IPCs in response to the activations of neuromodulatory neurons. The authors also conducted the connectome analysis and G-protein prediction analysis to strengthen their hypothesis that the heterogeneity of IPCs may underlie the flexible insulin release in response to various environmental conditions.Strengths:The authors succeeded patch-clamp recordings and calcium imaging of individual IPCs in living animals at a single-cell resolution, which allows them to show the heterogeneity of IPCs precisely. They measured IPC activities in response to 9 types of neurons in patch-clamp recordings and 5 types of neurons in calcium imaging, comparing the similarities and differences in activities between two methods. These results support the idea that the neuromodulatory system affects individual IPC activities differently in a receptor-dependent manner.

We thank the reviewer for emphasizing how our in vivo experiments allow for a precise characterization of the IPC responses to modulatory inputs.

Weaknesses:One concern is how much extent the heterogeneity of IPC activities in a short time scale is relevant to the net output, a release of insulin-like peptides in response to metabolic demands in a relatively longer time scale. The authors can test their hypothesis by manipulating the heterogeneous expressions of receptor genes in IPCs and examining IPC activities on a longer time scale. Moreover, while the authors focus on IPC activities, they did not show the activation of the neuromodulatory inputs and the net output of insulin levels in the data. The readers might want to know which neurons are indeed activated to send signals to IPCs and how IPC activities result in the secretion of insulin peptides.

We agree with the reviewer that the two experiments described, manipulating receptor expression before long-term recordings and measuring insulin levels after activating modulatory inputs, would deliver exciting insights into the interplay of modulatory inputs, IPC population activity, and insulin release. However, currently available methods for monitoring insulin release do not allow us to perform these experiments with a temporal resolution that would match the sensitivity and time resolution of our physiological experiments and are therefore not suited for a direct comparison. We also acknowledge that it would be extremely exciting to characterize the modulatory populations providing input to IPCs in terms of their sensitivity to internal state changes and external inputs. However, this clearly goes beyond the scope of our study. Essentially, one would have to perform experiments on a similar scale and breadth as we have done for IPCs here for the other populations. We aim to perform some of these experiments in follow up projects to this work.

**Reviewer #1 (Recommendations for the authors):**
(1) The authors used a 5% expression cutoff initially, which seems arbitrary. Can you explain the rationale for using this cutoff? If I interpret the authors' logic correctly and given there are 14 IPCs per animal, at 5% there is a 70% chance that 1 cell expresses that receptor.

We used a 5% cutoff to reduce false positives in our transcriptomic analysis. This threshold translates to expression in 0.8 out of 16 IPCs found in an individual fly on average. Hence, this cutoff ensures that receptors are expressed in at least 1 cell. Based on 392 IPC transcriptomes used in our analysis, our 5% threshold means that any receptor expressed in less than 20 transcriptomes will be deemed to be absent. At the population level, this ensures that our expression analysis is based on cells from at least two flies. However, we expect the actual number of flies from which the IPC transcriptomes were derived from to be much higher. We added the following statement to the methods section to clarify this point: “To determine if a transcript is present in the IPC transcriptomes, we used a 5% cutoff to reduce false positives. This cutoff is equivalent to expression in 0.8 IPCs out of 16 on average in an individual fly, and hence less than one IPC in the entire population. Since we used 392 IPC transcriptomes in our analysis, this cutoff means that expression in less than 20 IPCs will be deemed false positive”

(2) Were male and female brains examined separately and tested for divergent expression of T2A-reporter signals? While there were not many strong differences in the snRNAseq dataset, based on some discrepancies with the reporters it might be worthwhile to assess sex-specific differences that might account for the observed expression/non-expression of some receptors.

We did not investigate sex-specific differences using anatomical mapping, since our scRNA analysis pointed against that being a major factor. We clarified our reasoning in the results section by adding “Since there were little differences in receptor expression between males and females (Fig. S1C), we used the transcriptomes from both sexes for all subsequent analyses.” in the transcriptome section, and “Since baseline recordings from IPCs, in addition to our transcriptomic analysis, revealed no significant difference between male and female flies(26), we only used mated females for our physiological experiments.” in the transition to the physiology section of our manuscript.

(3) The anatomical reporter and transcriptome data for neuromodulatory receptor expression do not fully complement each other, e.g. in Fig1D Lkr is expressed only in one cluster but anatomical expression is observed in most IPCs. Ultimately, visualizing receptor expression at the protein level and functional analysis with genetic perturbation of the respective receptors is needed to draw strong conclusions.

We agree with the reviewer that visualizing receptor expression at protein level could help clarify some of these differences since neuropeptide GPCR transcripts tend to be less abundant whereas we expect protein expression to be more stable. However, out of the 14 receptors examined in our study, antibodies are only available for two: DH31R and LKR. Since our DH31R-T2A-GAL4 line does not drive expression in IPCs, we did not pursue this further. We did perform preliminary experiments to validate LKR protein expression in IPCs. Unfortunately, we found that the LKR antibody labels cells in the pars intercerebralis in both the wild type and LKR mutants (see Author response image 1 below). Therefore, we do not think it suitable to monitor LKR protein expression. Thus, additional investigations must await future generations of neuropeptide receptor antibodies. One biological reason for the discrepancies could be that anatomical quantification is based on cumulative expression while transcriptomic analysis captures a brief snapshot. We included “One explanation for the discrepancies could be that transcriptomic analysis provides a single snapshot, whereas anatomical data is based on cumulative expression. Fluorescent markers persist long after transcription and translation has terminated. Therefore, a higher likelihood for receptor expression can be expected when it is quantified via anatomical techniques.” in our results part to give the readers more context.

(4) In Fig1E, As Dop2R reporter signal is not colocalizing with IPC whereas dop2R is expressed in all four clusters.

We tested if additional transcript variants with different C-termini are the cause for the discrepancy between transcriptome data and anatomical mapping. However, using a Trojan-GAL4 line for Octa2R that should account for other transcript variants did also not show any expression. At this point, with the tools we have, we cannot conclusively determine what the cause of this discrepancy is. Since we only see them with Dop2R and Octa2R, a mismatch caused by more general differences,

e.g. sex-specific differences, seems unlikely. A more plausible reason could be that for those lines, inadequate transgenes lead to failed expressions. We added “Hence, inadequate transgenes for Dop2R and Octα2R or the lack of protein translation are the likely cause for the discrepancy between transcriptome analysis and anatomical mapping.“ to our results part as a possible explanation for the discrepancy.

(5) Moving the AstANs expression images to the main figure (Fig 1E) would make sense as the authors focus on AstAN rather than MsRT or Dop2R in the later parts of their work.

We thank the reviewer for this suggestion and replaced the LKR image with an AstAR2 image, as suggested. We kept the other two receptors in the main figure as additional examples.

(6) Have the authors considered gap junction coupling of IPCs, which might explain the simultaneous responses in some cases?

We have indeed considered this exciting idea, as gap junctions between IPCs could potentially synchronize activity in connected IPC subpopulations. To test if gap junctions are a major factor in the IPC population, we performed experiments with patch-clamp recordings from a single IPC while performing calcium imaging of the IPC population (as demonstrated in Fig. 4J). In some of these experiments, we injected current into individual IPCs and tested for activity changes in the other IPCs. However, the preliminary data we acquired did not indicate that the current-induced train of action potentials was transmitted to others IPCs. Hence, it is unlikely that the IPCs are directly coupled by gap junctions. Given the challenging nature of these experiments, and the discouraging preliminary results, we have not followed up on the idea any further.

**Reviewer #2 (Recommendations for the authors):**
(1) Figure 3D was not described in the text.

We thank the reviewer for pointing out this mistake, we included the panel in Figure 3C and added the reference in the text describing the results from multiple animals shown in the panel.

(2) In Figure 4B, a scale of heat map is required. There is a blue spot with no ROI setting on the left side. On the right side of the photos, the ROI No.6 seemed to turn blue after activation. However, Figure 4D shows the ROI No.6 was inhibited.

We are now using a simplified heatmap in Figure 4B and added a scalebar. We also changed the example images to avoid any confusion. Previously, we used a random snapshot from before LED onset, now we used a snapshot from the actual time window to which we normalized the traces. Regarding the spot where no ROI is depicted but a response is visible: in this area, a trachea made it difficult to clearly delimit the cell body underneath, and we therefore excluded this ROI. Occlusions by trachea are one reason why we can typically not image the entire IPC population in a single animal.

(3) In Figure 4F, the regions of gray bars (baseline) contain blue and red colors to some extent, which makes me confused. Moreover, the description "within one cluster, the response seemed homogeneous, e.g., in fly #4 during the activation of DANs (Fig. 4F)." was not clear to me. How about fly #1, #2, and #3? It seems that the responses changed excitedly and inhibitory within a cluster. Although the authors tend to raise some consistent results with examples, it would not be so effective if I can see there are other counter-examples and exceptions in the results.

We apologize for the confusion we caused. The gray bars indicate the time window we used for baseline subtraction: The median activity of each IPC in this window was subtracted from the activity of that IPC. Hence, the median activity in this window is zero, but individual frames can have positive or negative values.

We thank the reviewer for pointing out the confusion about the homogeneous responses in one cluster. We clarified this part in the results, by adding “Recording from multiple IPCs at the same time uncovered that the activity of IPCs within a cluster was synchronized in some cases. For example, in fly #1 in the DAN activation experiment, the baseline activity pattern of the excited IPC cluster was already synchronized before the first activation (fly #1, cells 3-8). Furthermore, the excitation onset and duration during the activation of DANs was highly uniform in this cluster. However, in other flies, e.g. #2 and #3 in the DAN activation experiments, we did not observe this synchronicity. While all IPCs in the excited cluster displayed an excitatory response to the DAN activation in these flies, the onset and duration differed between individual IPCs. In addition, the IPCs also showed more variability in their baseline activity (Fig. 4F). These findings point towards a shared input that can lead to the synchronization of IPC activity in some clusters and time windows. One known such input is the behavioral state – flight strongly inhibits the activity of all IPCs with very short delays(22). The flies in our experiments were not flying, but this example illustrates the presence of strong, state-dependent inputs that can synchronize the IPC population activity.”

(4) In Figure 4J, no explanations of arrowheads, gray boxes, or asterisks are available in the legend.

We thank the reviewer for pointing out this omission. We added the missing information to the figure legend.

(5) "IPCs form distinct clusters." Is this cluster located closely each other or distant from one another?

We did not encounter a location-dependent relationship between the IPCs of one cluster in calcium imaging experiments, nor did the anatomical receptor mapping data or connectomics analysis give any indication for anatomical clusters. The location of individual IPC cell bodies is not stereotypical across flies. We clarified this point in the results by adding “IPCs form distinct functional clusters” and “However, we found no evidence in our anatomical data, calcium imaging experiments, or in the fly brain EM volume that these clusters are distinguishable based on IPC soma location in the pars intercerebralis.”